



# Molecular insights on aging and aqueous phase processing from ambient biomass burning emissions-influenced Po Valley fog and aerosol

Matthew Brege[1], Marco Paglione[2], Stefania Gilardoni[2], Stefano Decesari[2], Maria Cristina Facchini[2], and Lynn R. Mazzoleni[1,3]

[1]Department of Chemistry, Michigan Technological University, Houghton, MI, USA.
[2]Institute of Atmospheric Sciences and Climate, Italian National Research Council, Bologna, Italy.
[3]Atmospheric Sciences Program, Michigan Technological University, Houghton, MI, USA.

*Correspondence to:* Lynn R. Mazzoleni (lrmazzol@mtu.edu)

**Abstract.** Atmospheric organic matter is a complex mixture of thousands of individual organic compounds originating from a combination of primary emissions and secondary processes. To study the influence of regional biomass burning emissions and secondary processes, ambient samples of fog and aerosol were collected in the Po Valley (Italy) during the 2013 Supersito field campaign. After the extent of "fresh" *vs.* "aged" biomass burning influence was estimated from proton nuclear magnetic resonance ($^1$H-NMR) and high resolution time of flight aerosol mass spectrometry (HR-ToF-AMS) observations, two samples of fog water and two samples of PM$_1$ aerosol were selected for ultrahigh resolution Fourier transform ion cyclotron resonance mass spectrometry (FT-ICR MS) analysis. Over 4300 distinct molecular formulas were assigned to electrospray ionization FT-ICR MS anions and were sorted into four elemental groups (CHO, CHNO, CHOS and CHNOS) and 64 subclasses. Molecular weight distributions indicated that the water-soluble organic matter was largely non-polymeric without clearly repeating units, although some evidence of dimerization was observed for C$_{10}$ compounds and especially for C$_{8-9}$ CHNO species in the "aged" aerosol. The selected samples had an atypically large frequency of molecular formulas containing nitrogen and sulfur (not evident in the NMR composition) attributed to multifunctional organonitrates and organosulfates. While higher numbers of organonitrates were observed in aerosol (dry or deliquesced particles), higher numbers of organosulfates were mostly found in fog water, and so chemical reactions promoted by liquid water must be postulated for their formation. Consistent with the observation of an enhanced aromatic proton signature in the $^1$H-NMR analysis, the average molecular formula double bond equivalents and carbon numbers were higher in the "fresh" biomass burning influenced samples, whereas the average O:C and H:C values from FT-ICR MS were higher in the samples with an "aged" influence (O:C > 0.6 and H:C > 1.2). The "aged" fog had a large set of unique highly oxygenated CHO fragments in HR-ToF-AMS mass spectra, which reflects an enrichment of carboxylic acids and other compounds carrying acyl groups as highlighted by the NMR analysis. Fog compositions were more "SOA-like" than aerosols as indicated by the observed similarity between the aged aerosol and fresh fog, implying that fog nuclei must be somewhat aged. Overall, functionalization with nitrate and sulfate moieties, in addition to aqueous oxidation, trigger an increase in the molecular complexity in this environment, which is apparent in the FT-ICR MS results. This study demonstrates the significance of the aqueous phase to transform the molecular chemistry of atmospheric organic matter and contribute to secondary organic aerosol.

## 1. Introduction

Atmospheric organic aerosol particles are comprised of a complex mixture of numerous individual organic compounds, produced by direct emissions and evolving secondary processes, of which a significant impact is from transformations in the aqueous phase. Surface emitted primary organic aerosol and volatile organic compounds are transformed in the atmosphere by gas to particle phase conversion, heterogeneous reactions, and aqueous phase reactions in aerosol water, fog, and cloud droplets (Ervens



et al., 2011; Herrmann et al., 2015). The products of these processes are collectively referred to as secondary organic aerosol (SOA). These aging reactions happen quickly in the atmosphere, and the observed mass fraction of SOA is larger than that of primary organic aerosol (Zhang et al., 2007; Ervens et al., 2011; Zhang et al., 2011; Paglione et al., 2014; Gilardoni et al., 2016). Biomass burning emissions, such as those from agricultural land clearing, forest fires, residential heating, and cooking with
biofuels, are important sources of organic carbon to the atmosphere globally (Andreae and Merlet, 2001; Bond et al., 2004; Glasius et al., 2006; Laskin et al., 2015). Biomass burning products include simple organic acids and dicarboxylic acids, sugars and anhydrosugars, substituted phenols, polycyclic aromatic hydrocarbons, and more, depending on the type of fuel and burn conditions (Mazzoleni et al., 2007; Pietrogrande et al., 2014a; Pietrogrande et al., 2014b; Gilardoni et al., 2016). These water-soluble emissions can serve as precursors for SOA once dissolved in the aqueous phase (Chang and Thompson, 2010; Yu et al., 2014; Yu
et al., 2016), and upwards of 50% of organic matter in fog and cloud droplets remains unidentified (Herckes et al., 2013). Biomass burning emissions can even facilitate droplet nucleation. In fact, laboratory studies indicate that, in addition to hydrophilic species, even refractory "tar balls," emitted from smoldering biomass burning begin to absorb water at high relative humidity (Hand et al., 2005; Laskin et al., 2015).

Due to their complexity, SOA concentrations, chemical oxidation states, functionalization, and high molecular weight
compounds, such as atmospheric humic-like substances (Ervens et al., 2011; Lee et al., 2013; Nguyen et al., 2013), are poorly represented in atmospheric chemistry models. Aqueous phase reactions in wet aerosol, cloud, and fog droplets have been proposed to improve the SOA observation gaps (Ervens et al., 2011; Herckes et al., 2013; Laskin et al., 2015; Gilardoni et al., 2016), but the current level of understanding regarding aqueous phase processes is insufficient to include them in models. Laboratory studies focusing on simplified systems of only one or two precursor components have successfully recreated some of the complexity of
ambient atmospheric samples (De Haan et al., 2011; Lee et al., 2013; Nguyen et al., 2013; Hawkins et al., 2016; Yu et al., 2016). A number of recent studies focusing on the molecular composition of cloud (Lee et al., 2012; Desyaterik et al., 2013; Pratt et al., 2013; Zhao et al., 2013; Boone et al., 2015; Cook et al., 2017) and fog (Mazzoleni et al., 2010; LeClair et al., 2012; Xu et al., 2017) chemistry have been recently reported. Together these studies indicate a clear importance of aqueous phase reactions for the production of aqueous SOA, including the formation of organonitrates, organosulfates, and nitrooxy-organosulfates. Of these,
organosulfate formation is thought to happen nearly exclusively in the aqueous phase (Ervens et al., 2011; Herrmann et al., 2015). Along with organonitrates, organosulfates are susceptible to hydrolysis in the aqueous phase, though high kinetic barriers under atmospheric conditions often slow these reactions and allow for the observation of these species in ambient samples (Darer et al., 2011; Hu et al., 2011). Organosulfates are often described in the literature as the products of acid catalyzed oxidation of biogenic terpenoids (Surratt et al., 2008; Pratt et al., 2013; Schindelka et al., 2013), but have also been observed in biomass combustion
influenced cloud water (Zhao et al., 2013; Cook et al., 2017). The formation of aqueous phase products in aerosol, fog and cloud waters, greatly increase the complexity of organic aerosol. Although several analytical techniques have been used to address the challenge of resolving the complex mixture of atmospheric organic matter (Decesari et al., 2007; Hertkorn et al., 2007; Nizkorodov et al., 2011; Desyaterik et al., 2013; Dall'Osto et al., 2015; Noziere et al., 2015; Laskin et al., 2016; Willoughby et al., 2016), no universal analytical method exists.
Ambient conditions ideal for observations of the aqueous phase contributions to atmospheric organic matter exist in the Po Valley (Italy). Surrounded by mountains to the north, west and south, the valley frequently has stable meteorological conditions with low ventilation and a low boundary layer, allowing for the accumulation of high concentrations of regional pollutants. The valley contains a mixture of dense population areas and intensively cultivated agricultural regions. Consequently, frequent fog events and high concentrations of anthropogenic biomass burning emissions are observed in months with cold temperatures (Larsen
et al., 2012; Saarikoski et al., 2012; Giulianelli et al., 2014; Paglione et al., 2014; Gilardoni et al., 2016). The Po Valley has some





of the highest reported carbon concentrations for fog water in the world (Herckes et al., 2013). In recent years, the analysis of fog water and aerosol from San Pietro Capofiume (SPC; located 30 km northeast of Bologna) has included Aerodyne high resolution time-of-flight aerosol mass spectrometry (HR-ToF-AMS) and proton nuclear magnetic resonance spectroscopy ($^1$H-NMR) to determine the fog scavenging efficiency of aerosol (Gilardoni et al., 2014) and source apportionment of aerosol (Decesari et al.,

2007). In Saarikoski et al. (2012), HR-ToF-AMS data from SPC aerosol showed an extremely high concentration of aerosol nitrate (39%) and a somewhat typical concentration of organic carbon (33%) in agreement with Gilardoni et al. (2014). Positive matrix factorization (PMF) of HR-ToF-AMS organic mass fragments was used to identify several factors describing Po Valley organic aerosol, including factors for fresh biomass burning organic aerosol, and three types of oxygenated organic aerosol. A similar study by Paglione et al. (2014) used PMF on $^1$H-NMR data of SPC aerosol to identify factors for fresh biomass burning emissions, as

well as SOA factors, including products formed from aged biomass burning emissions.

Further investigation with a focus on molecular markers and source apportionment was done as part of the Supersito 2013 field campaign in the Emilia-Romagna region, including samples from SPC and the urban site of Bologna (Pietrogrande et al., 2014a; Pietrogrande et al., 2014b; Poluzzi et al., 2015). The campaign has shown the significance of biomass burning emissions in the region. Approximately 35% of the organic carbon was from wood burning in winter months (Pietrogrande et al., 2015),

biomass burning emissions were shown to increase with decreasing ambient temperature (Gilardoni et al., 2014), and aqueous phase SOA formation from biomass burning emissions and associated brown carbon formation was directly observed (Gilardoni et al., 2016). HR-ToF-AMS observations have shown similarity between atmospheric organic matter in fog water and aerosol formed following fog dissipation, indicating low volatility organics that were originally present in the fog are left behind upon evaporation; these particles are enriched in oxidized organic matter, and absorb solar radiation more efficiently than fresh

emissions, contributing to atmospheric brown carbon (Gilardoni et al., 2016).

In this study, we analyzed fog from SPC and aerosol from Bologna, collected during the 2013 Supersito field campaign. Due to the intense time investment required for FT-ICR MS data analysis, we chose to focus our detailed analysis on a subset of samples, including two aerosol and two fog samples. The subset was selected to represent the influence of fresh and aged biomass burning emissions on fog and aerosol based on the HR-ToF-AMS and $^1$H-NMR observations (Section 3.1). We used the

combination of $^1$H-NMR, HR-ToF-AMS and FT-ICR MS techniques, which revealed detailed chemical information about the complex mixtures of atmospheric organic matter in the Po Valley. Molecular insights on aging and aqueous phase processing were gathered from the variations in the detailed molecular composition from ultrahigh resolution FT-ICR MS measurements within the chemical context provided by the $^1$H-NMR, HR-ToF-AMS, and other measurements. Similar studies focusing on analysis of atmospheric samples with $^1$H-NMR and FT-ICR MS have been conducted in the past (Schmitt-Kopplin et al., 2010; Willoughby

et al., 2016), but so far, this type of study with a focus on biomass burning and aqueous phase processing has not been previously reported.

## 2. Methods

### 2.1 Sample collection and chemical analysis

Sub-micrometer (PM$_1$) aerosol particles were collected in Bologna on pre-washed and pre-baked quartz-fiber filters

(PALL, 18 cm diameter) by a High-Volume Sampler (TECORA Echo Hi Vol) equipped with a digital PM$_1$ sampling inlet, at a nominal flow rate of 500 L min$^{-1}$. PM$_1$ samples were collected during winter 2013 (from 4 February 2013 to 15 February 2013), in the framework of the Supersito project. Fog water was collected at the SPC field station, where monitoring of fog occurrence and fog water collection has been performed every year from November to March systematically since 1989 (Giulianelli et al.,



2014); during the 2013 winter, more than 20 fog samples were collected from 29 November 2012 to 12 March 2013. In the fog collector (Fuzzi et al., 1997), a short wind tunnel is created by a rear fan, where an air stream containing fog droplets are collected by impaction using a series of stainless steel strings. The collected droplets drain off the strings into a sampling bottle. The air flow through the tunnel was 17 $m^3 min^{-1}$ with a 50% collection efficiency for individual strings (3 μm radius each). All parts of the fog collector coming into contact with the fog droplets, were made of stainless steel to avoid sampling artifacts from adsorption of organic compounds to the surfaces.

The aerosol filters were extracted with deionized ultra-pure water (Milli-Q) in an ultrasonic bath for 1 h. The water extract was filtered with a 0.45 μm PTFE membrane in order to remove suspended particles. Fog water was filtered through 47 mm quartz fiber filters within a few hours of collection and conductivity and pH measurements were taken (Crison microCM 2201 conductimeter and Crison micropH 2002 pH meter). Aliquots of both aerosol water extracts and fog water were used to determine the total organic carbon content (Multi N/C 2100 analyzer; Analytik Jena, Germany) and water soluble organic carbon (WSOC) concentration (Rinaldi et al., 2007).

### 2.2 $^1$H-NMR analysis

Aliquots of the aerosol extract and fog water were dried under vacuum and re-dissolved in deuterium oxide (D$_2$O) for organic functional groups characterization by $^1$H-NMR spectroscopy, as described in Decesari et al. (2000). The $^1$H-NMR spectra were acquired at 600 MHz (Varian Unity INOVA spectrometer) with a 5 mm probe. Sodium 3-trimethylsilyl-(2,2,3,3-d4) propionate (TSP-d4) was used as an internal standard by adding 50 μl of a 0.05% TSP-d4 (by weight) in D$_2$O to the standard in the probe. The speciation of hydrogen atoms bound to carbon atoms can be provided by $^1$H-NMR spectroscopy in protic solvents, on the basis of the range of frequency shifts, the signal can be attributed to H-C containing specific functionalities (Decesari et al., 2000; Decesari et al., 2007). For more than a decade, $^1$H-NMR has been used for the characterization of organic aerosol. Detection limits for an average sampling volume of 500 $m^3$ were of the order of 3 nmol $m^{-3}$ for each functional group. A total of 21 fog and 18 aerosol $^1$H-NMR spectra were collected during the winter 2013 campaign; the method described above was used to identify and quantify major components of WSOC. In the present study, the results of these $^1$H-NMR analyses were used to characterize and to select the samples for subsequent FT-ICR MS analysis as described in section 3.1.

### 2.3 HR-ToF-AMS analysis

During the Supersito winter 2013 campaign (4 February 2013 to 15 February 2013) the chemical composition of submicron aerosol particles at Bologna was characterized with a 5 minute-time resolution using an HR-ToF-AMS (Aerodyne Research (DeCarlo et al., 2006)). Data was collected in the V-ion mode, at a resolution of 2,200. The influx of aerosol particles was dried below 30% relative humidity with a Nafion drier before analysis. Details on analysis of HR-ToF-AMS data for the Supersito winter 2013 campaign were previously reported (Gilardoni et al., 2016); here we report HR-ToF-AMS characterization averaged over the sampling periods of the selected aerosol samples.

Fog water samples were also analyzed by HR-ToF-AMS after being re-aerosolized (TSI constant output atomizer, Model #3076) in an inert argon gas flow, to characterize dissolved water-soluble organics. To make sure that the re-aerosolized fog water represented the original sample, we verified that the nitrate-to-organic carbon and the sulfate-to-organic carbon ratios from the HR-ToF-AMS analysis were within 20% (measurement uncertainty level) of the ratios measured off-line by ion chromatography and thermo-optical analysis.



### 2.4 Ultrahigh resolution FT-ICR MS analysis

Four samples were selected for FT-ICR MS analysis based on the characterization by HR-ToF-AMS data and [1]H-NMR spectra for the entire Supersito winter 2013 sample set (Section 3.1). High molecular weight WSOC compounds were prepared for FT-ICR MS analysis using a polymeric reversed phase solid phase extraction (SPE) cartridge (Strata-X, Phenomenex) to remove

salts and low molecular weight compounds which interfere with electrospray ionization (ESI). The cartridges were loaded with HCl acidified aqueous samples (pH < 2), rinsed with 1 mL of water, and then eluted using 2 mL of ACN:$H_2O$ (90:10 by volume). Fog samples were filtered with a quartz filter before SPE. A portion of the aerosol filter samples were extracted with ultrapure water using sonication and the extracts were then filtered using a quartz filter to remove insoluble materials; the aerosol extracts were then prepared for FT-ICR MS analysis using SPE as described above. The WSOC described in this paper is operationally

defined as the WSOC that is both retained and recovered from the SPE cartridges (SPE-recovered), thus it is not equivalent with the total WSOC. The ACN:$H_2O$ extracts were analyzed at the Woods Hole Oceanographic Institute in Woods Hole, MA, using full-scan ESI ultrahigh-resolution FT-ICR MS (7T LTQ FT-ICR MS, Thermo Scientific) at a resolving power of 400,000 as described in our previous work (Zhao et al., 2013; Dzepina et al., 2015). We used direct infusion analysis to collect mass spectrometry data over the mass range of *m/z* 100-1000 in the negative ionization mode, for approximately 200 scans. Molecular

formulas were assigned as previously described in our work (Mazzoleni et al., 2010; Putman et al., 2012; Zhao et al., 2013; Dzepina et al., 2015) using Sierra Analytics Composer software (version 1.0.5) within the limits of $C_{2-200}H_{4-1000}O_{1-20}N_{0-3}S_{0-1}$. The formulas were reviewed manually for their credibility; for further details, see the Supplemental Text. Approximately 74% of the measured masses in each of the samples were assigned a molecular formula. Oxygen to carbon (O:C) and hydrogen to carbon (H:C) ratios, were calculated from the respective number of C, H or O atoms in the assigned molecular formulas. We calculated Kendrick mass

(KM) and Kendrick mass defect (KMD) as described in equations (1) and (2), respectively (Stenson et al., 2003).

$$KM = experimental\ mass * \left(\frac{14.00000}{14.01565}\right) \qquad (1)$$

$$KMD = nominal\ mass - KM \qquad (2)$$

DBE was calculated by equation (3) for the molecular formula format: $C_cH_hO_oN_nS_s$.

$$DBE = c - \left(\frac{h}{2}\right) + \left(\frac{n}{2}\right) + 1 \qquad (3)$$

Note that S and O are divalent in equation (3); additional bonds formed by tetravalent and hexavalent S are not included in the DBE calculations. The average oxidation state of carbon ($OS_C$) in the molecular formulas was calculated using equation (4) as described in Kroll et al. (2011) and the modified aromaticity index ($AI_{mod}$) (Koch and Dittmar, 2006, 2016) was calculated using equations (5-7) with the same molecular formula format as DBE.

$$OS_C = 2 * (O:C) - (H:C) \qquad (4)$$

$$DBE_{AI} = 1 + c - \left(\frac{o}{2}\right) - s - \left(\frac{n+h}{2}\right) \qquad (5)$$

$$C_{AI} = c - \left(\frac{o}{2}\right) - n - s \qquad (6)$$

$$AI_{mod} = \frac{DBE_{AI}}{C_{AI}} \qquad (7)$$

In equation (7), the $AI_{mod} = 0$, if $DBE_{AI} \leq 0$ or $C_{AI} \leq 0$, as defined in Koch and Dittmar (2006, 2016).



### 3. Results and discussion

#### 3.1 Selection of aerosol and fog water samples

Among the 15 fog and 18 aerosol samples collected during the winter of 2013 at SPC and Bologna, we selected two fog and two aerosol samples for subsequent analysis by FT-ICR MS. The aerosol samples were selected to represent conditions strongly impacted by "fresh" and "aged" wood burning emissions based on the evaluation of the $PM_1$ organic aerosol composition in the winter using HR-ToF-AMS and $^1$H-NMR data as described in Gilardoni et al. (2016). Using PMF source apportionment, we observed the highest concentration of SOA between 11 February 2013 and 20 February 2013. In particular, on 13 February 2013, the ratio of SOA to POA was ~4 and the aqueous SOA from biomass burning accounted for about 55% of the total SOA. Thus, this aerosol sample (BO0213D) was strongly influenced by aged wood burning emissions. During the night of 4 February 2013, the fresh biomass burning concentration was ~6 µg m$^{-3}$, accounting for the 54% of total organic aerosol; thus, it was strongly influenced by fresh wood burning emissions. Similarly, HR-ToF-AMS observations were used to select fog samples strongly impacted by "fresh" and "aged" wood burning emissions. Specifically, we used the relative intensity of $m/z$ 60 ($f_{60}$) as a marker of fresh biomass burning influence and $m/z$ 44 ($f_{44}$) as a marker of oxygenated and processed dissolved organic molecules (Aiken et al., 2008; Gilardoni et al., 2016). The $f_{44}$ $vs.$ $f_{60}$ space was previously proposed to represent biomass burning $vs.$ atmospheric aerosol aging (Cubison et al., 2011) and was extended here to fog samples. In Fig. 1a, it can be seen that the fog sample SPC0106F had low $f_{44}$ and high $f_{60}$ values, while SPC0201F had high $f_{44}$ and low $f_{60}$ values. Thus, from here on, SPC0106F (fog) and BO0204N (aerosol) will be referred to as the "fresh" biomass burning influenced samples, and SPC0201F (fog) and BO0213D (aerosol) will be referred to as the "aged" biomass burning influenced samples. A summary of the sample collection details and HR-ToF-AMS characterization is given in Table 1.

#### 3.2 $^1$H-NMR composition

Functional group distributions for the selected $PM_1$ and fog samples were provided by $^1$H-NMR analysis. A synthetic representation of the $^1$H-NMR organic functional groups distribution of all the collected samples is reported in Fig. 1b, following the approach described by Decesari et al. (2007) for source attribution. Briefly, Decesari et al. (2007) presented a survey of $^1$H-NMR functional group distributions of WSOC samples from diverse environments proposing fingerprints for broad categories of oxygenated organic compounds in aerosol, namely SOA (enriched in acyl groups, H-C-C=O), biomass burning aerosol (enriched in alkoxyls, H-C-O, and aromatics), and marine organic aerosol (rich of aliphatic groups other than acyls and alkoxyls, mainly amines and sulfoxy groups). In this study, most samples were categorized either as SOA or biomass burning, even if a significant fraction of the aerosol samples exhibited $^1$H-NMR compositions with a very high alkoxyl contribution, exceeding the boundaries proposed by Decesari et al. (2007). For example, sample BO0204N (representative of fresh biomass burning aerosol) showed by far the largest contribution of alkoxyl groups and the least amount of acyl groups. In contrast, BO0213D (representative of aged aerosols) showed relatively high acyl content and small alkoxyl fractions. Similarly, the two selected fog samples (SPC0106F: fresh, and SPC0201F: aged) were clearly differentiated based on their $^1$H-NMR functional group distributions (Fig 1b). Therefore, the selected aerosol and fog samples represent extremes in the structural space of this WSOC sample set based on the distribution of $^1$H-NMR functionalities, and in agreement to the categorization provided by the HR-ToF-AMS measurements.

The differences between the two aerosol samples likely reflect the ambient conditions during sampling: BO0204N was characterized by night-time accumulation of ground-level local emissions from residential heating and an absence of photochemical processes; instead, BO0213D was characterized by daytime photochemically processed aerosol and by an enhanced mixing with regional-scale air masses. Similarly, the diversity in the fog samples reflects the collection duration and the associated





liquid water content (LWC) of the two considered fog events: SPC0106F was collected over a shorter duration with a lower LWC compared to SPC0201F (Table 1).

It should be noted that although a pair of fresh and aged samples were selected from each of the sample sets, Fig. 1 shows a clear shift in the average composition between the fog and the aerosol samples, where the fog samples were characterized by a
greater amount of acyl groups and a smaller fraction of alkoxyls. So, according to the simple source-attribution scheme based on the major [1]H-NMR functionalities presented here, the fog compositions were more "SOA-like" than aerosols. As a consequence, the fresh fog composition overlapped with the aged aerosol composition (Fig. 1b). This implies that the fresh fog sample SPC0106F was processed to a similar degree as the most aged aerosol sample BO0213D. This was confirmed by the corresponding HR-ToF-AMS elemental ratios (very similar O:C for SPC0106F and BO0213D, see Table 1) and by the detailed comparison between the
[1]H-NMR spectra of these two samples (one fog and one aerosol). This difference in the average functional group composition between fog and aerosol samples in the Po Valley can be explained by: (a) the preferential scavenging of more oxidized constituents of organic particles into fog (Gilardoni et al., 2014); (b) the effect of oxidative chemical reactions in fog water leading to the production of carboxylic acids and carbonyls (hence, acyls); and (c) a stronger aging effect from fog processing at the rural site (SPC) with respect to urban areas (Bologna) at the margins of the Po basin.

The [1]H-NMR spectra of the selected samples are reported in Fig. 2. The spectra of the aerosol samples (Fig. 2c and 2d) exhibited a clear biomass burning fingerprint, with evident proton resonances from levoglucosan and intense bands from alkoxyl (H-C-O) and aromatic (Ar-H) groups. However, the band of phenols and methoxyphenols, which are primary biomass burning tracers, were clearly found only in the spectrum of BO0204N, representative of fresh primary organic aerosols in our study. Moreover, the fraction of levoglucosan and alkoxyl groups was much greater in BO0204N than in BO0213D. The aged aerosol
BO0213D contained higher amounts of two methylamines (mono- and tri-methyl-amines) relative to BO0204N, and especially much larger fractions of methanesulfonate and succinic acid, which are tracers of SOA. The spectral region between the chemical shift of 2.1 and 2.4 ppm showed clear bands representing aliphatic dicarboxylic acids and ketoacids (Suzuki et al., 2001) in the aged aerosol, but were barely visible in the fresh aerosol. The aged aerosol was also characterized by the occurrence of hydroxy-methanesulfonic acid (HMSA), a known tracer of aqueous SOA. Similarly, SPC0106F (Fig. 2A) exhibited a clear biomass burning
fingerprint with contributions from levoglucosan, alkoxyl (H-C-O), and aromatic groups (Ar-H), whereas SPC0201F (Fig. 2B) showed tracers of aqueous-phase SOA (HMSA) and high concentrations of acyl groups, (CH-C=O), which demonstrated the effects of the aging process. Additionally, SPC0201F exhibited several low-molecular weight organic acids (phthalic, maleic, succinic, pyruvic and lactic acids) in much greater amounts than SPC0106F, where only traces of phthalic and succinic acids were found. This indicated that the aged fog was enriched in products of the oxidative degradation of particulate and gaseous organic
compounds. It should be noted, that the fresh fog did not show the prominent band from phenols or methoxyphenols observed in the spectrum of the fresh aerosol (BO0204N). This suggests that the WSOC of the fresh fog had undergone a certain degree of chemical modification respective to primary biomass burning OA, even if the secondary products of such transformations were not visible in the [1]H-NMR spectrum.

**3.3 Ultrahigh resolution FT-ICR MS composition**

**3.3.1 Overview of the Po Valley ambient fog and aerosol compositions**

Approximately 1600-2800 individual monoisotopic molecular formulas were assigned to the ultrahigh resolution mass spectra of the SPE-recovered WSOC from each Po Valley sample. Based on the inclusion of C, H, N, O, and S elements, the molecular formulas were sorted into the following elemental groups: "CHO," "CHNO," "CHOS" and "CHNOS." The percent composition of these elemental groups for each sample is shown in Fig. 3. Most of the molecular formulas were present in the





subclasses $O_{4-10}$, $NO_{3-13}$, $O_{5-10}S$ and $NO_{7-11}S$ (Fig. S1). A summary of the observed numbers of formulas per elemental group, as well as the average O:C, H:C, $OS_C$ and DBE values are provided in Table 2. Although they are not expected to match, the values for the SPE-recovered WSOC do trend with those from the HR-ToF-AMS data shown in Table 1; we note, that not only are the elemental ratios from different fractions of the aerosol, but they are also determined differently.

5        A great diversity of organonitrates, organosulfates, and nitrooxy-organosulfates were observed in the Po Valley samples. These compound classes can be inferred from the O:N and O:S of the assigned molecular formulas. Nearly all N-containing formulas had O:N > 3, suggesting that a majority of the nitrogen species contained at least one nitrooxy (nitrate) group. Multiple nitrogen species, such as those of classes $N_2O_{3-5}$ and $N_3O_{5-7}$ have an O:N low enough to make imine and imidazole structures possible, and these types of products have been reported in cloud water mimic reactions (De Haan et al., 2011). All of the S-

containing formulas had O:S > 4 ratios, indicating sulfite, sulfate, and sulfonic acid functionalities. These inferences are consistent with the ionization polarity, where oxidized and acidic components are more efficiently ionized in negative ion ESI. A study by LeClair et al. (2012) who performed FT-ICR MS/MS on a variety of CHNO, CHOS, and CHNOS components confirmed that the studied compounds in Fresno fog were indeed multifunctional organonitrates, organosulfates, and nitrooxy-organosulfates. Furthermore, nitrate and sulfate salts are common secondary components present in the Po Valley (Giulianelli et al., 2014) and

reactions between these inorganic salts and organics are expected as secondary reactions in the aqueous phase (Noziere et al., 2010; McNeill et al., 2012; Herrmann et al., 2015; McNeill, 2015). Amines have been observed in the Po Valley, emitted by livestock farming and waste treatment activity, and it is possible that some species with amine groups were emitted from smoldering biomass combustion (Andreae and Merlet, 2001; Paglione et al., 2014). However, given the analytical bias for acidic functional groups in the ESI negative ion mode, it is unlikely that reduced nitrogen species were detected. Nitrated phenols are known contributors to

light absorbing atmospheric brown carbon and are associated with biomass burning (Desyaterik et al., 2013; Laskin et al., 2015). In this work, a large number of CHNO formulas were observed with low H:C and low O:C, especially in the fresh aerosol and fresh fog samples (Fig. S2); several of the CHNO formulas were also estimated to be aromatic using the $AI_{mod}$ calculation (Fig. 4). Specifically, the molecular formulas for 4-nitrophenol, 2-methyl-4-nitrophenol, 2,4-dinitrophenol, 4-nitroguaiacol and 3-nitrosalicylic acid (Kitanovski et al., 2012; Desyaterik et al., 2013) were observed in all four Po Valley samples.

25        All of the molecular formulas were plotted in van Krevelen space (H:C *vs*. O:C) partitioned by sample (columns) and elemental group (rows; Figs. 5, S2). In this space, molecular formulas with O:C ≥ 0.6 and $OS_C$ ≥ 0 are considered to be highly oxidized and formulas with H:C ≥ 1.2 are considered to be highly saturated (Tu et al., 2016). The distribution of the CHO and CHNO formulas is quite similar to WSOC extracted from ambient fog collected in Fresno, CA (Mazzoleni et al., 2010). Additionally, the distribution of CHO formulas from phenolic aqueous SOA reported in Yu et al. (2016) partially covers the same

area of the van Krevelen space. The CHOS and CHNOS formulas with high H:C ratios were also distributed similarly to Mazzoleni et al. (2010). The high H:C ratios indicate that a majority of the CHOS and CHNOS formulas represent aliphatic organosulfate compounds, consistent with the aliphatic $AI_{mod}$ values (Fig. 4). In contrast, a majority of the formulas with aromatic $AI_{mod}$ values were in the CHO and CHNO groups, and tended to cluster at low H:C and low O:C in the van Krevelen space, in agreement with previous studies (Mazzoleni et al., 2010; LeClair et al., 2012). Consistent with the [1]H-NMR results in Fig. 1B, the van Krevelen

diagrams for SPC0106F and BO0213D were similar (see also Fig. S2), barring the additional low H:C CHOS and CHNOS formulas of SPC0106F and the additional CHNO formulas of BO0213D.

       Underscoring the influence of biomass burning on these samples, we found several molecular formulas matching previously observed species in biomass burning influenced ambient cloud water from Mt. Tai, China (Desyaterik et al., 2013). There were also several matches with the products of laboratory phenolic aqueous SOA reactions (Yu et al., 2014; Yu et al., 2016)

(Table 3 and Table S1). Other notable molecular formulas included those for the compounds: acetosyringone, acetovanillone,


azelaic acid, benzoic acid, coumaryc acid, hydroxybenzoicacid, ketolimononaldehyde, methyl-nitrophenol, nitrocatechol, nitrophenol, o-toluic acid, phthalic acid, syringaldehyde, syringic acid, tyrosine, vanillic acid and vanillin (Table 3) (Mazzoleni et al., 2007; Desyaterik et al., 2013; Nguyen et al., 2013; Pietrograndhe et al., 2014a; Pietrograndhe et al., 2014b; Yu et al., 2014; Dzepina et al., 2015; Pietrograndhe et al., 2015; Yu et al., 2016). The molecular formulas for common methoxyphenols (syringol

($C_8H_{10}O_3$), methylsyringol ($C_9H_{12}O_3$), and eugenol ($C_{10}H_{12}O_2$)) were present in all samples except BO0204N; as they are both semi-volatile and water-soluble, they are not expected to be present in aerosol with low liquid water content. Several formulas were also found that could be more oxidized versions of phenolic species produced from biomass burning. These formulas included additional oxygen atoms added to the base formulas for phenol ($C_6H_6O_{3-5}$), guaiacol ($C_7H_8O_{3-6}$) and syringol ($C_8H_{10}O_{4-7}$). Five of these formulas, $C_6H_6O_3$, $C_6H_6O_5$, $C_8H_{10}O_5$, $C_8H_{10}O_6$ and $C_8H_{10}O_7$, were previously observed in biomass burning aerosol

(Pietrograndhe et al., 2015) and in the products of laboratory phenolic aqueous phase SOA reactions (Yu et al., 2014; Yu et al., 2016).

### 3.3.2 Molecular trends for ambient fog and aerosol compositions

Molecular formula trends in the form of histograms are a useful way to organize and visualize the thousands of molecular formulas observed here. The trends based on carbon number, oxygen number, and DBE of the assigned molecular formulas are

shown in Fig. 6. Although relative abundance does not directly correspond to analyte concentrations, it provides a basis for relative comparisons. For example, the influence of terpene SOA products is indicated from the elevated total relative abundance of molecular formulas near $C_{10}$ (observed in all samples) and an additional increased abundance between $C_{15-18}$ (observed in most samples). This was especially pronounced in BO0213D (Fig. 6a). These formulas are likely derived from monoterpenes ($C_{10}$) and sesquiterpenes ($C_{15}$), where terpene emissions have been observed in biomass burning (Andreae and Merlet, 2001). Terpene

oxidation products, including organosulfates, were previously observed in biomass burning influenced cloud water (Zhao et al., 2013; Cook et al., 2017) and many of the same molecular formulas were observed in this study (Table S1). Specifically, we observed molecular formulas for pinic acid, ketopinic acid, pinonic acid, 3-hydroxy-4,4-dimethylglutaric acid, and 3-methyl-1,2,3-butanetricarboxylic acid (Table 3) (He et al., 2014). Overall, the trends indicate an enhanced abundance of CHO formulas in SPC0201F, CHNO formulas in BO0204N and BO0213D, and CHOS formulas in SPC0106F (Fig 6a). Consistent with the [1]H-

NMR results in Fig. 1b, there is a strong similarity between samples SPC0106F and BO0213D, especially for the oxygen and DBE trends shown in Figs. 6b and 6c.

Difference mass spectra were constructed from the assigned monoisotopic molecular formulas for the fog and aerosol samples (Fig. S3) and provide a direct comparison of their compositions. Each of the individual relative abundances were normalized by the total abundance of the assigned masses for each sample. In Fig. S3, the individual masses with higher abundances

in either the positive or negative direction were substantially greater in the fresh or aged samples, respectively; the masses with similar relative abundances tended to cancel each other. Overall, we observed higher numbers of oxygen in the molecular formulas at lower molecular weights in the two aged samples compared to the two fresh samples. To investigate this further, we adapted the approach used for the molecular formula trends described above with the difference relative abundances. The resulting difference trend plots are shown in Fig. 7 for carbon, and Figs. S4 and S5 for oxygen and DBE respectively. In Fig. 7b, it is clear there was

an enhanced abundance of CHOS and CHNOS formulas with higher carbon numbers in the fresh fog, while the aged fog showed an enhanced abundance of low carbon number CHO formulas. In Fig. 7a, it is clear that the fresh aerosol had an enhanced abundance of higher carbon number formulas, though unlike the fog samples, they were mainly CHO and CHNO compounds. The aged aerosol had an enhanced abundance of low carbon number formulas from the CHOS and CHNOS groups. In both fog and aerosol, there is an enhanced abundance of higher carbon numbers in the fresh samples relative to the aged samples. Overall, the



carbon numbers are shifted to lower values in the fog compared to aerosol (Fig. 7) and the oxygen numbers are shifted to higher values in fog compared to aerosol (Fig. S4).

The subsequent sections discuss the molecular diversity of the different samples, especially considering the sample type, atmospheric processes during sample collection and the unique molecular formulas observed in each sample. The distributions of
unique molecular formulas are shown in Fig. 3b and Fig. S1.

### 3.3.3 Comparison of the fresh and aged biomass burning influenced fog compositions

As expected from the molecular trends, the molecular formulas of the aged biomass burning influenced fog (SPC0201F) were more oxidized than the fresh biomass burning influenced fog (SPC0106F). The overall average O:C was higher in SPC0201F than in SPC0106F across all groups. Likewise, the average number of oxygen in CHO and CHNO formulas were higher in the
aged fog, however the average number of oxygen in CHOS and CHNOS formulas was about the same. Fig. S4a shows this enhancement in oxidation with a greater abundance of higher oxygen number formulas observed in the aged fog. In the van Krevelen space, the molecular formulas in SPC0106F tended to cluster to the left of the O:C = 0.6 line in Fig. 5, but the formulas in SPC0201F tended to cluster on the line and slightly to the right of it, indicating they were significantly more oxygenated. Additionally, there were more molecular formulas in SPC0106F below the H:C = 1.2 line (Fig. 5 and Fig. S2), indicating these
formulas were more unsaturated. The average $OS_C$ was higher across all groups in SPC0201F compared to SPC0106F, underscoring the more oxidized nature of the aged fog; this can be seen in Fig. 5 as additional formulas in SPC0201F were shifted towards and below the $OS_C$ = 0 diagonal line. The average DBE values for all of the elemental groups were higher in SPC0106F compared to SPC0201F (Table 3); similarly, the average $AI_{mod}$ was higher, indicating there are more aromatic structures in SPC0106F. Most of the CHOS and CHNOS formulas in SPC0106F and SPC0201F were classified as aliphatic by $AI_{mod}$ and
approximately 30% of these formulas in SPC0106F were classified as olefinic, which was higher than any other sample. In addition to the higher average DBE and $AI_{mod}$, the average number of carbon atoms in the SPC0106F molecular formulas was higher as well. This suggests that the fresh fog molecular formulas represented molecules with large unsaturated carbon backbones, which is consistent with pollutants without significant atmospheric aging. In contrast, the molecular formulas with smaller carbon backbones that were more oxidized were more prevalent in the aged fog.

The unique molecular formulas found in SPC0106F were mostly of the $O_{5-13}S$ and $NO_{7-12}S$ subclasses. Since organosulfates are associated with aqueous secondary processes (Darer et al., 2011; Ervens et al., 2011; Schindelka et al., 2013; McNeill, 2015), the fact that 33% of the formulas in SPC0106F contained sulfur was initially confusing. Likewise, non-aromatic organosulfates and nitooxy-organosulfates would not be expected in fresh biomass burning emissions (Staudt et al., 2014). Furthermore, photolysis reactions were not expected to have influenced the sample composition because it was collected from
~3:10 am to 4:30 am. However, the fog composition is influenced by the preceding fog nuclei composition (Herckes et al., 2007; Gilardoni et al., 2014), (Darer et al., 2011; Hu et al., 2011) indicating that aged aerosol act as fog nuclei. This is further underscored by the similar compositions of the fresh fog and aged aerosol samples. Meanwhile it is well known that organosulfates are products of aqueous-phase SOA reactions, which are enhanced at acidic pH, (Noziere et al., 2010; Ervens et al., 2011; McNeill et al., 2012). Thus, the lower LWC and relatively higher solute concentrations may have promoted CHOS and CHNOS compound formation in
the fresh fog. Increasing organosulfate concentrations in aerosol have been shown to be linked to increased aerosol LWC (Huang et al., 2015), although it is unclear if this correlation extends to fog (as described in Gilardoni et al. (2014), fog drops have LWC ≥ 8 x $10^4$ µg m$^{-3}$). A noticeable number of CHOS and CHNOS formulas unique to SPC0106F had higher DBE values than formulas from other samples. There was an observed preference for a significant fraction of CHOS and CHNOS formulas with DBE values < 6, except for some of the formulas in SPC0106F (Fig. 6c). The 10 most abundant unique molecular formulas in the fresh biomass





burning influenced fog of SPC0106F were all CHOS and CHNOS formulas: $C_{15}H_{24}O_7S$, $C_{13}H_{14}O_8S$, $C_{14}H_{16}O_8S$, $C_{15}H_{24}O_8S$, $C_{18}H_{30}O_8S$, $C_{15}H_{16}O_9S$, $C_{16}H_{18}O_9S$, $C_{17}H_{20}O_9S$, $C_{19}H_{24}O_9S$ and $C_{11}H_{15}NO_8S$. These formulas may be tracer species for partially fog processed biomass burning emissions.

While all samples contained some unique molecular formulas among the CHO subclasses, a high number of formulas in the $O_{9-14}$ subclasses were unique to the aged fog (SPC0201F). This trend could indicate enhanced oxidation and aging as a result of aqueous phase reactions in fog. The high average O:C ratio ($0.577 \pm 0.18$) and low pH (3.34) of SPC0201F is consistent with the trend observed by Cook et al. (2017) for cloud water, where the average O:C increased with decreasing pH. Overall, we observed a significant number of CHNO, CHOS and CHNOS molecular formulas in SPC0201F, which are expected products of secondary aqueous phase reactions in fog. However, there was a lower percentage of CHNO, CHOS and CHNOS formulas, and

an increased percentage of CHO formulas in SPC0201F compared to SPC0106F, suggesting that aqueous SOA products with N or S may have been transformed by acid hydrolysis into more stable CHO species (Darer et al., 2011). The increased oxidation is supported by the [1]H-NMR analysis, which showed an enrichment of carboxylic acids and other compounds carrying acyl groups. It is possible that an enhanced oxidation from hydroxyl radicals and photolysis reactions played a role in the composition of this sample, as sunrise was ~3.5 hours before the sample collection ended. SPC0201F had additional unique formulas which were

highly oxygenated in the $NO_{13}$, $O_{11}S$ and $NO_{13}S$ subclasses, which appeared on the low mass end of the homologous series in the CHOS and CHNOS groups (Fig. S6). The 10 most abundant unique molecular formulas in the aged biomass burning influenced fog (SPC0201F) were CHO, CHOS and CHNOS species with smaller carbon skeletons than the fresh biomass burning influenced fog (SPC0106F), including: $C_{10}H_{10}O_7$, $C_{11}H_8O_7$, $C_{12}H_{14}O_9$, $C_{10}H_{18}O_5S$, $C_8H_{14}O_7S$, $C_8H_{12}O_7S$, $C_9H_{16}O_8S$, $C_4H_9NO_7S$, $C_5H_9NO_7S$, and $C_8H_{13}NO_{11}S$. These formulas may be tracer species for heavily fog processed biomass burning emissions.

**3.3.4 Comparison of the fresh and aged biomass burning influenced aerosol compositions**

Consistent with the fog samples, the average O:C and H:C values, and average number of oxygen atoms, were higher in the aged aerosol (BO0213D) than in the fresh aerosol (BO0204N). The influence of fresh biomass burning emissions is further supported by the higher average DBE, $AI_{mod}$, and average number of carbon atoms in BO0204N. These trends were clearly apparent for the CHO and CHNO groups, but less apparent for the CHOS and CHNOS groups. However, a low number of S-containing

formulas were observed in BO0204N (~6%), which may have skewed these statistics. CHO and CHNO formulas in BO0204N clustered to the left of the O:C = 0.6 line in Fig. 5, indicating they were not very oxygenated. In contrast, the aged aerosol molecular formulas were distributed on and to the right of the O:C = 0.6 line. Additionally, there were more formulas above the H:C = 1.2 line in BO0213D compared to BO0204N. The distribution of CHOS and CHNOS formulas in the van Krevelen space was similar for BO0204N and BO0213D, however there was a greater number of formulas in the aged aerosol BO0213D.

Like the fog samples, more formulas that represent molecules with larger carbon backbones and higher DBE were found in the fresh aerosol sample compared to the aged aerosol sample, and molecular formulas that represent molecules with smaller carbon backbones that were more oxidized and oxygenated were comparatively more prevalent in the aged aerosol sample. These carbon number and DBE trends are clearly visible through the difference trends shown in Figs. 7a and S6a, respectively. Both aerosol samples had a high percentage of formulas that contained nitrogen, with a noticeable number of CHNO formulas unique

to the samples (Fig. 3b). This larger percentage of CHNO formulas may be attributed enhanced $NO_X$ concentrations associated with urban traffic emissions (Glasius et al., 2006) (Table 1). However, residential wood combustion influenced cloud water collected near Steamboat Springs, CO, was found to be composed of ~52% CHNO molecular formulas (Zhao et al., 2013), and elevated numbers of CHNO formulas were also reported in aerosol with a strong regional biomass burning influence (Schmitt-Kopplin et al., 2010) and wildfire influenced cloud water (Cook et al., 2017).



A majority of the unique formulas in the fresh aerosol (BO0204N) were in the $NO_{6-12}$ and $N_2O_{7-11}$ subclasses. These unique CHNO formulas were less saturated and less oxygenated, compared to the formulas in the aged aerosol (BO0213D). In BO0204N, these formulas were expected to be products of $NO_X$ reactions and night-time nitrate radical reactions. Since BO0204N was collected during a period with low relative humidity and had low aerosol liquid water, it seems unlikely that these products

were formed by aqueous phase secondary processes. The low humidity also explains the low percentage of CHOS and CHNOS formulas observed in BO0204N. Overall, no unique CHOS formulas were detected in BO0204N and only 1 unique CHNOS formula ($C_8H_{11}NO_8S$) was detected. The small number of observed non-unique CHOS and CHNOS formulas in BO0204N may have originated from the increase in LWC (up to ~300 µg m$^{-3}$) observed in the last 4 hours of sample collection, and thus may have been formed by similar processes as in BO0213D. Less oxygenated species would be expected with little to no influence

from aqueous phase secondary processes (Ervens et al., 2011; McNeill, 2015), such as observed in the low humidity and low aerosol liquid water conditions of the fresh aerosol in BO0204N. We also observed an overall decrease in the relative abundances of molecular formulas with DBE ≥ 5 in all samples except for BO0204N, where there was a secondary spike near DBE 10 (Fig. 6c). The 10 most abundant unique molecular formulas in the aerosol with a fresh biomass burning influence (BO0204N) were mostly $N_1$ and $N_2$ CHNO formulas: $C_{16}H_{15}NO_6$, $C_{24}H_{23}NO_{10}$, $C_{24}H_{21}NO_{10}$, $C_{26}H_{23}NO_{10}$, $C_8H_4N_2O_6$, $C_{12}H_{10}N_2O_8$, $C_{13}H_{12}N_2O_8$,

$C_{15}H_{14}N_2O_{10}$, $C_{17}H_{20}O_5$ and $C_{20}H_{18}O_8$. These formulas may be tracer species for biomass burning emissions when night-time gas phase reactions are dominant.

Several unique molecular formulas for the aged aerosol in BO0213D were found in the $N_2O_{4-13}$ and $N_3O_{5-13}$ subclasses, as well as the $O_{4-7}S$, $NO_{5-7}S$ and $NO_{10-12}S$ subclasses. A large fraction of the $N_2$ formulas, and all of the $N_3$ formulas were unique to BO0213D. Compared to the other samples, BO0213D was collected during relatively high $NO_X$ conditions, as well as high

humidity and aerosol liquid water content. Furthermore, this was the only sample collected during daylight hours, implying that these $N_2$ and $N_3$ formulas are related to photochemical reactions. The increased frequency of CHOS and CHNOS formulas in BO0213D compared to BO0204N was likely from the formation of organosulfates and nitrooxy-organosulfates in the aqueous phase, enhanced by the increased concentration of species in aerosol liquid water (Darer et al., 2011; Hu et al., 2011; McNeill et al., 2012). Accretion reactions such as aldol condensation, acetal, and hemiacetal reactions are also expected to take place at a

significant rate in these enhanced concentrations (Herrmann et al., 2015). While there was not a significant trend towards higher masses in BO0213D compared to other samples, the unique molecular formulas of this sample tended to fall on the high mass end of the homologous series, especially for CHNOS formulas (Fig. S6). The 10 most abundant unique molecular formulas for BO0213D were mostly highly oxygenated CHNO formulas: $C_7H_9NO_3$, $C_{16}H_{18}N_2O_{11}$, $C_{16}H_{18}N_2O_{11}$, $C_{17}H_{22}N_2O_{11}$, $C_{18}H_{24}N_2O_{11}$, $C_{17}H_{22}N_2O_{13}$, $C_{18}H_{21}N_3O_{11}$, $C_9H_{15}NO_{10}$, $C_{12}H_{25}NO_8S$ and $C_{15}H_{24}O_{12}$. These formulas may be tracer species for biomass burning

emissions heavily aged by reactions in aerosol liquid water with photolysis.

## 4. Summary and implications

Hydrophilic species are expected to enhance droplet formation, indicating that organics acting as fog nuclei must be somewhat aged. In fog or wet aerosol, the water-soluble organics are subjected to further transformation in the aqueous phase, as we have observed here. These transformation processes in fog and aerosol water were shown to produce oxygenated and oxidized

molecular formulas, as well as N-containing and S-containing formulas with what were likely nitrate and sulfate functional groups. On the basis of the analysis of the selected aerosol and fog samples, representing extreme cases in the HR-ToF-AMS and $^1$H-NMR projections of the organic aerosol structural space, we can summarize the following observations:



- An overall molecular trend was observed for both fog and aerosol samples, of concurrent shifts from lower H:C and O:C in samples with fresh biomass burning influence, and toward higher H:C and O:C values in samples with aged biomass burning influence. This was consistent with the [1]H-NMR functional group distributions, which showed a decrease of aromatic moieties from the fresh to the aged aerosol, largely due to the disappearance of phenolic structures. Aged sample molecular formulas contained on average a higher number of oxygen atoms, while fresh sample molecular formulas contained on average a higher number of carbon atoms. The lower number of carbon atoms observed in aged samples suggests that the secondary formation of oligomers was somewhat counterbalanced by fragmentation reactions and/or by the uptake of low-molecular weight compounds from the gas-phase. However, some evidence of dimerization of $C_{10}$ compounds was found in all samples, especially for $C_8$-$C_9$ CHNO compounds in the aged aerosol. Overall, the fog composition was generally more "SOA-like" than the aerosol, where the fresh fog composition was similar to the aged aerosol composition in both the [1]H-NMR analysis and the molecular formula trends.

- CHOS and CHNOS formulas were detected with high frequencies in samples that were collected as activated fog (both samples) or had a substantial aerosol liquid water content during collection (as in the case of BO0213D). This provided strong evidence that the production of S-containing SOA species is dominated by reactions in the aqueous phase. The occurrence of S-containing SOA in the fog can be explained by formation in incipient droplets (at very low LWC), through radical reactions mediated by the sulfate radical (Noziere et al., 2010; McNeill et al., 2012; Schindelka et al., 2013), or possibly acid catalyzed substitution reactions with the sulfate ion (Darer et al., 2011; Hu et al., 2011; McNeill, 2015). When comparing the unique formulas of the two aged samples (SPC0201F and BO0213D), aging reactions in aerosol liquid water appeared to produce less highly oxygenated CHO formulas than in fog and a greater number of formulas in the CHNO, CHOS and CHNOS groups. This difference could be explained by the increased chance of reactions with inorganic nitrate and sulfate ions in the relatively higher solute concentrations of aerosol liquid water compared to the increased likelihood of hydration reactions in fog (Darer et al., 2011; Hu et al., 2011). This conclusion agrees with the quantitative analysis of functional group composition of aqueous SOA isolated by positive matrix factorization analysis of HR-ToF-AMS mass spectra and [1]H-NMR spectra reported previously (Gilardoni et al., 2016).

- Compared to fresh fog (SPC0106F), the aged fog (SPC0201F) had an enhancement in the highly oxidized CHO formulas and an overall lower percentage of CHNO and CHOS formulas. This is likely due to hydrolysis reactions in the low pH environment (Darer et al., 2011; Hu et al., 2011). The [1]H-NMR analysis also highlighted that SPC0201F included several low-molecular weight organic acids (phthalic, maleic, succinic, pyruvic acids) which originated from the degradation of particulate WSOC, the oxidation of condensable water-soluble volatile organic compounds, and the uptake of condensable products of gas-phase oxidative reactions. The resulting [1]H-NMR spectral fingerprint of the aged fog (SPC0201F) is clearly distinct from those of the fresh fog and the two aerosol samples, which are instead dominated by spectral features of primary biomass burning components. Overall, the variability of [1]H-NMR fingerprints between samples reflects the change in oxidation state of the CHO family detected by FT-ICR MS (reaching a maximum for SPC0201F), but seems rather insensitive to the changes in content of heteroatom-containing groups (CHNO, CHOS, CHNOS). In fact, the formation of CHOS compounds detected by ultrahigh resolution FT-ICR MS analysis in deliquesced aerosols or in low-LWC fog water (e.g., BO0213D and SPC0106F) could not be traced to parallel changes in [1]H-NMR spectral characteristics. It is possible, however, that a fraction of the [1]H-NMR–detected alkoxyl groups (H-C-O) were in fact bound to sulfate esters and misclassified as alcohols.

In this work, we used the detailed molecular composition to describe the differences in aging and aqueous phase processes for a select set of samples from the Supersito 2013 winter campaign. A majority of the molecular formulas observed in this study





have not been previously reported, but correlate with anticipated molecular trends. This emphasizes the importance of detailed molecular analysis of atmospheric samples for the study of biomass burning emissions processed in the aqueous phase of aerosol and fog, as well as the potential of aqueous phase processing to act as a source of SOA in the atmosphere.

**Data availability**

5 An abbreviated list of the complete FT-ICR MS dataset is provided and is available on Digital Commons: http://digitalcommons.mtu.edu/chemistry-fp/98/

**Acknowledgments**

This research was supported with a NASA Earth and Space Science Fellowship for Matthew Brege. The Italian CNR contribution was supported by the BACCHUS project, European Commission FP7-603445. The authors thank Drs. Melissa Soule and Elizabeth

10 Kujawinski of the Woods Hole Oceanographic Institution (WHOI) Mass Spectrometry Facility for instrument time and assistance with data FT-ICR MS acquisition (NSF OCE-0619608 and Gordon and Betty Moore Foundation).


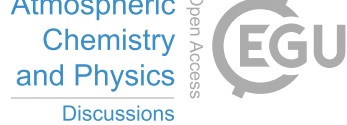

**Figures**

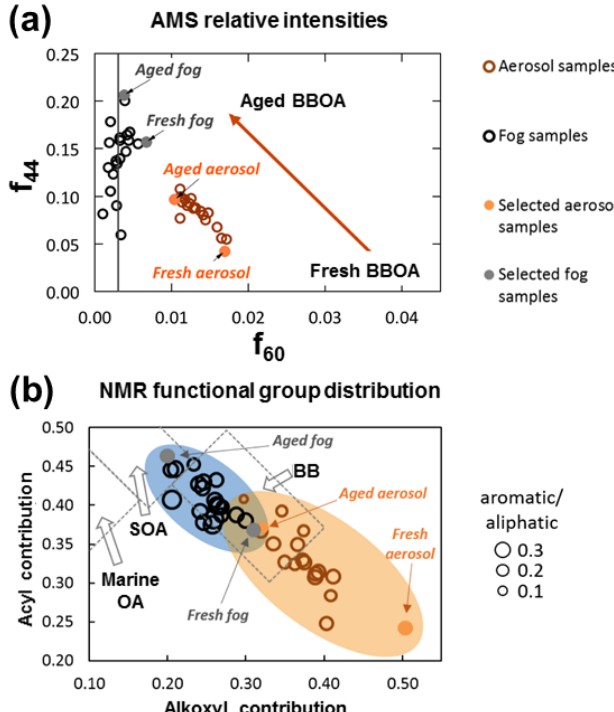

**Figure 1:** Preliminary characterization of fog and $PM_1$ aerosol samples collected in SPC and Bologna, respectively, during the 2013 Supersito field campaign. Characterization was performed via HR-ToF-AMS analysis as described by Cubison et al. (2011), utilizing the relative intensity of peak $m/z$ 60 ($f_{60}$) and peak $m/z$ 44 ($f_{44}$) as markers of fresh biomass burning influence and oxygenated and processed dissolved organic molecules respectively (a). Further characterization was performed via $^1$H-NMR analysis, as described by Decesari et al. (2007), where samples were mapped by $^1$H-NMR functional group fractions (b). In (b), dashed lines indicate the boundaries of the source fingerprints according to Decesari et al. (2007), ("BB": biomass burning aerosol) and the x and y axes report the contributions of alkoxyl (H-C-O) and acyl (H-C-C=O) groups to the total aliphatic fraction of WSOC respectively. The sample names Fresh Fog, Aged Fog, Fresh Aerosol, and Aged Aerosol correspond to SPC0106F, SPC0201F, BO0204N, and BO0213D, respectively.





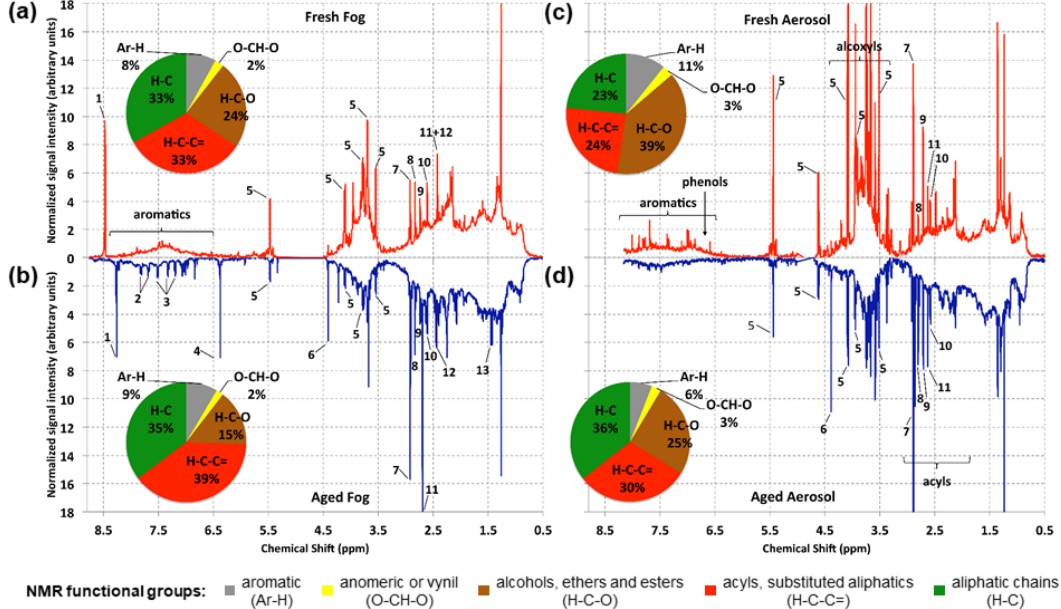

**Figure 2:** The $^1$H-NMR spectra of selected fog water (a and b) and aerosol (c and d) samples, and their corresponding functional groups distribution. A set of specific resonances was attributed to individual compounds: 1) formate, 2) phthalic acid, 3) ammonium, 4) maleic acid, 5) levoglucosan, 6) hydroxy-methanesulfonic acid, 7) trimethylamine, 8) methanesulfonic acid, 9) dimethylamine, 10) monomethylamine, 11) succinic acid, 12) pyruvic acid, 13) lactic acid. The sample names Fresh Fog, Aged Fog, Fresh Aerosol, and Aged Aerosol correspond to SPC0106F, SPC0201F, BO0204N, and BO0213D, respectively.



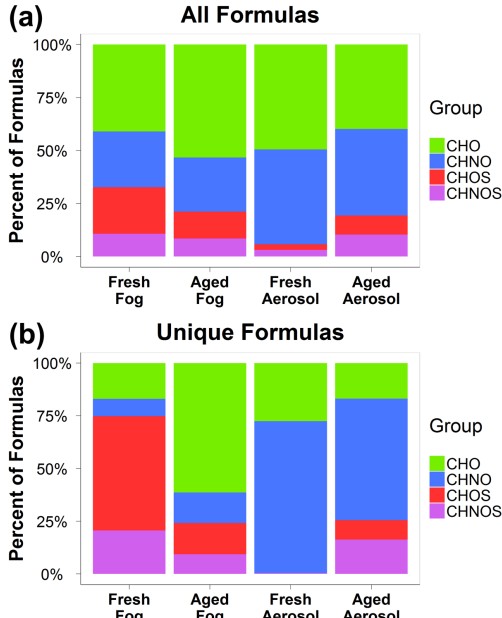

**Figure 3:** Percentage of assigned molecular formulas to each of the elemental groups in the Po Valley samples, where (a) includes all identified
molecular formulas and (b) includes only the unique molecular formulas. The sample names Fresh Fog, Aged Fog, Fresh Aerosol, and Aged
Aerosol correspond to SPC0106F, SPC0201F, BO0204N, and BO0213D, respectively.





**Figure 4:** The modified aromaticity index ($AI_{mod}$) for the assigned molecular formulas (Equations 5-7) and the percentage of each $AI_{mod}$ type, as defined by Koch and Dittmar (2016): aliphatic ($AI_{mod} = 0$), olefinic ($0 < AI_{mod} \leq 0.5$), aromatic ($AI_{mod} > 0.5$), and condensed aromatic ($AI_{mod} \geq 0.67$). Here aromatic and condensed aromatic formulas were combined, because a small fraction of condensed aromatics was observed. The results are partitioned by elemental group, where it can be seen that the majority of olefinic and aromatic compounds belong to the CHO and CHNO groups. The sample names Fresh Fog, Aged Fog, Fresh Aerosol, and Aged Aerosol correspond to SPC0106F, SPC0201F, BO0204N, and BO0213D, respectively.





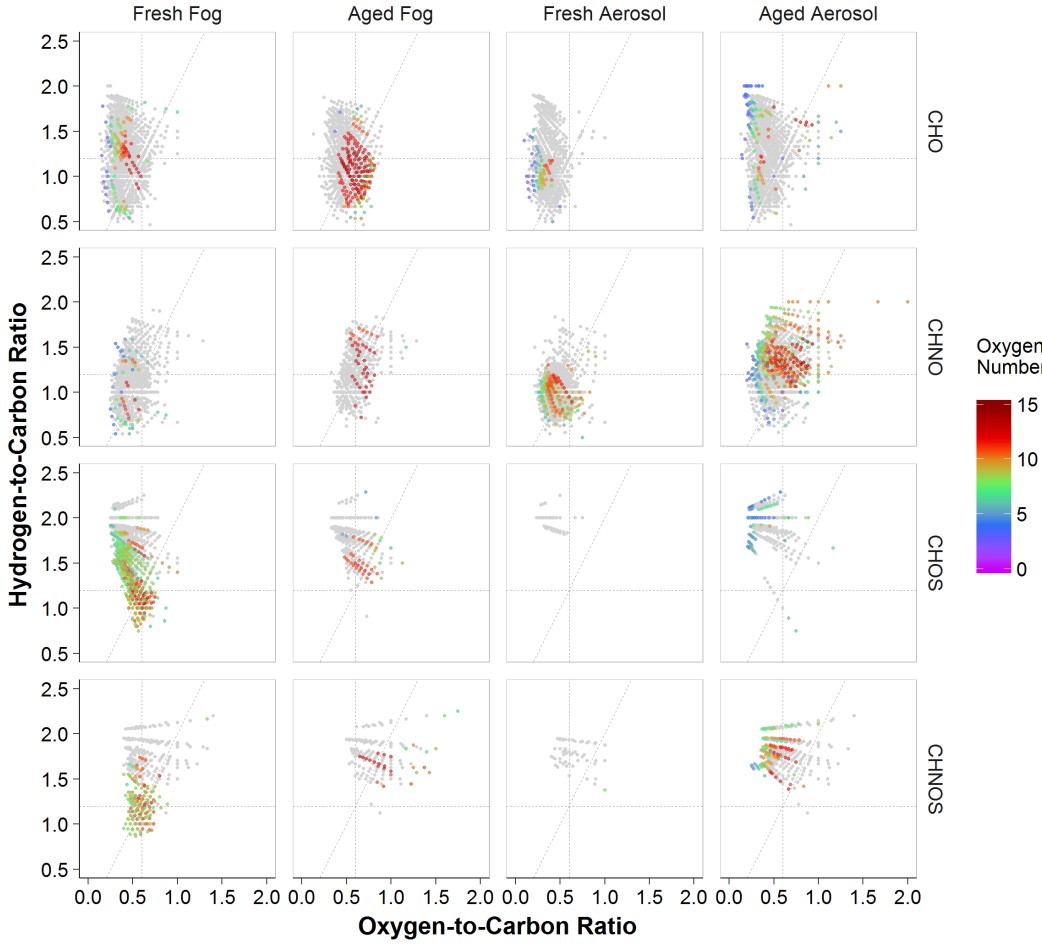

**Figure 5:** van Krevelen diagrams for the SPE-recovered WSOC by elemental group (rows) and sample (columns) as indicated in the Figure. Dashed lines represent H:C = 1.2 (horizontal), O:C = 0.6 (vertical) and $OS_C$ = 0 (diagonal) as described in Tu et al. (2016). Formulas unique to each sample are color scaled to the number of oxygen atoms in the assigned formula; grey points represent common molecular formula assignments. The sample names Fresh Fog, Aged Fog, Fresh Aerosol, and Aged Aerosol correspond to SPC0106F, SPC0201F, BO0204N, and BO0213D, respectively. A similar plot with all of the molecular formulas scaled to indicate the number of oxygen atoms is provided as Fig. S2.



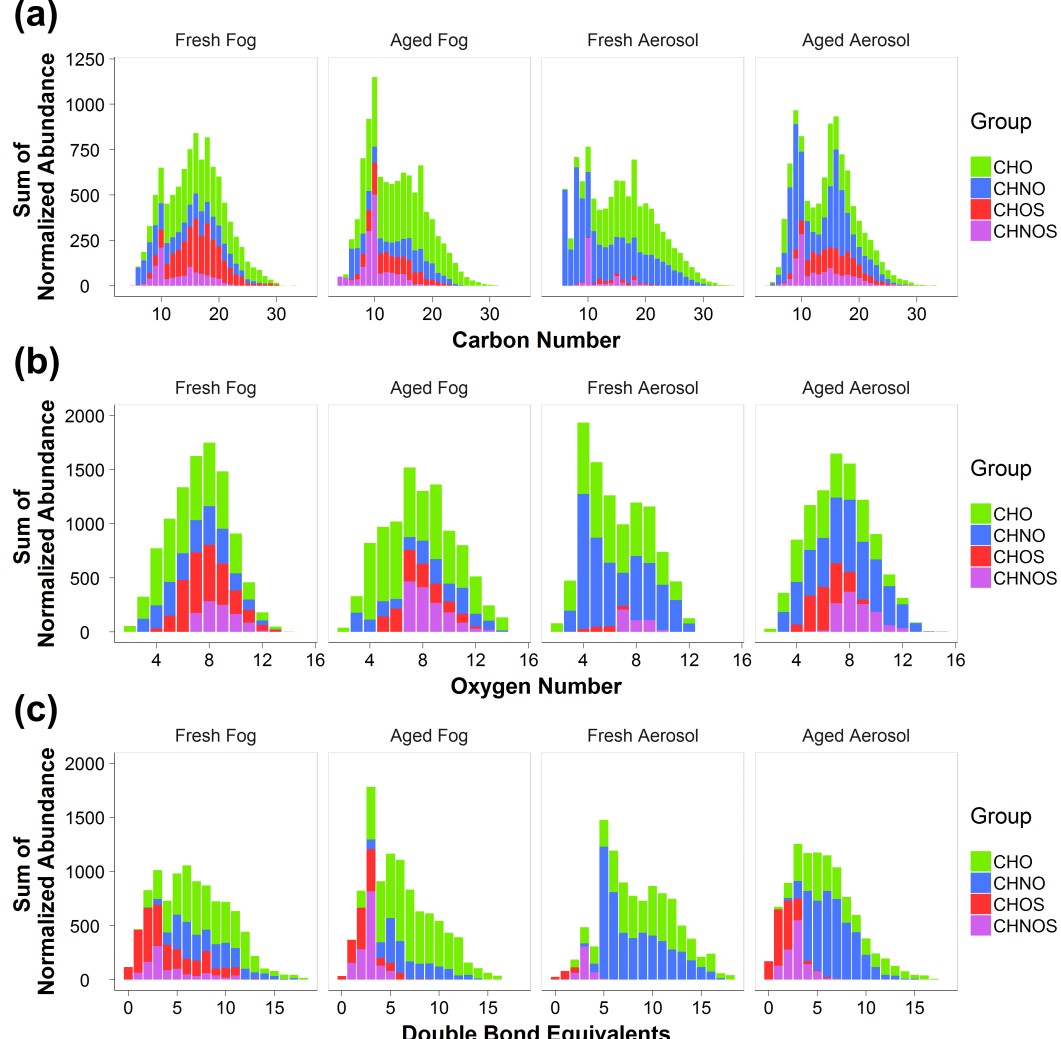

**Figure 6:** Molecular formula trends for carbon (a), oxygen (b) and the number of double bond equivalents (c). All detected molecular formula abundances were normalized to the total assigned ion abundance for each sample and then summed across the integer values for carbon number, oxygen number, or double bond equivalent values. The sample names Fresh Fog, Aged Fog, Fresh Aerosol, and Aged Aerosol correspond to SPC0106F, SPC0201F, BO0204N, and BO0213D, respectively.



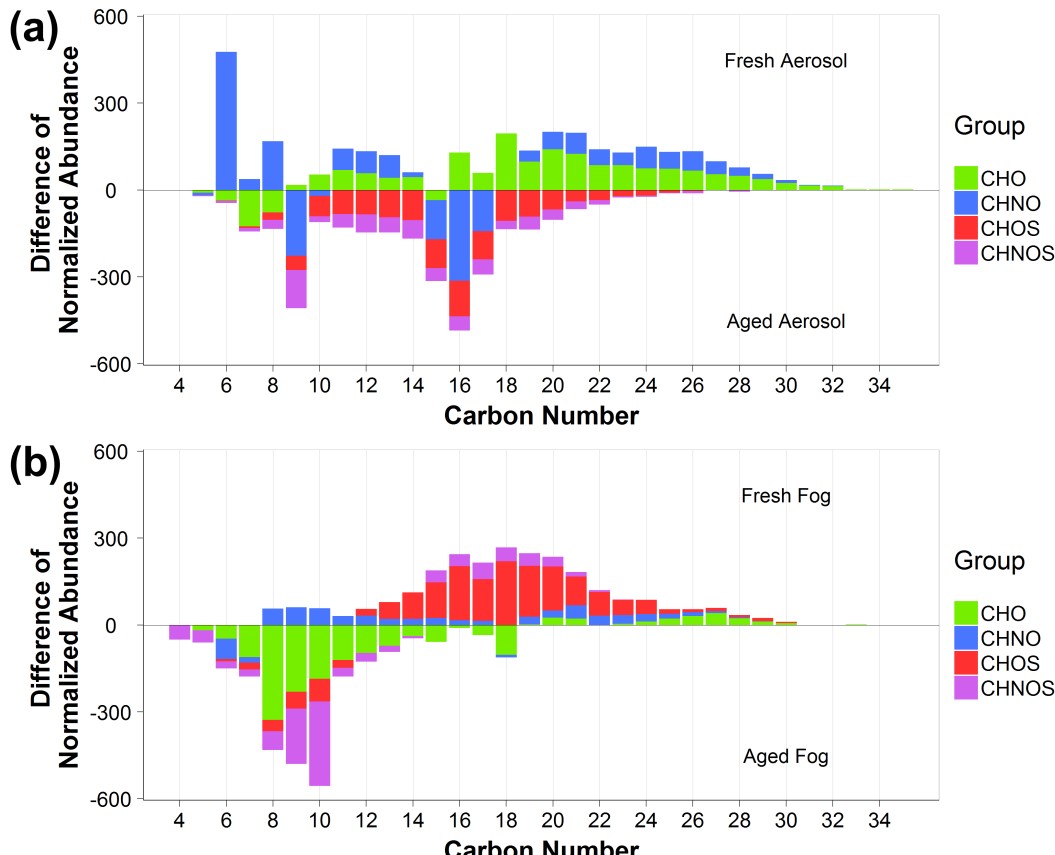

**Figure 7:** Carbon difference trend plots for aerosol (a) and fog (b) sample types. Difference trends were calculated as in Figure 6 and then the respective aged sample was subtracted from the fresh sample for each integer carbon number value. Positive values indicate an enhanced relative abundance of the formulas in the fresh sample compared to the aged sample. Similarly, negative values indicate an enhanced abundance of formulas in the aged sample compared to the fresh sample. The sample names Fresh Fog, Aged Fog, Fresh Aerosol, and Aged Aerosol correspond to SPC0106F, SPC0201F, BO0204N, and BO0213D, respectively.





**Tables**

**Table 1:** Sample collection, identification and HR-ToF-AMS data. Relative humidity (RH), liquid water content (LWC), and aerosol liquid water content (ALWC) are averaged over the sample collection time. Fog samples were collected at Capofiume (SPC). Fog water samples were re-aerosolized for HR-ToF-AMS data analysis, while aerosol sample data is from on-line measurements. For aerosol samples, the standard deviation of on-line measurements corresponding to the sample collection period are shown.

| Sample name | SPC01016F | SPC0201F | BO0204N | BO0213D |
|---|---|---|---|---|
| Collection site | SPC | SPC | Bologna | Bologna |
| Sample type | Fog water | Fog water | PM$_1$ aerosol | PM$_1$ aerosol |
| Fresh *vs.* Aged influence | Fresh | Aged | Fresh | Aged |
| Start collection date and time[a] | 6 January 2013, 3:10 | 1 February 2013, 19:40 | 4 February 2013, 18:18 | 13 February 2013, 9:24 |
| Collection time (h) | 1.33 | 15.37 | 14.62 | 8.60 |
| Temperature (°C)[b] | 1.0 | 3.0 | 5.9 | 3.0 |
| pH | 5.81 | 3.34 | NA | NA |
| [NO$_X$] (ppb)[b] | 73 | 15 | 146 | 101 |
| RH (%)[b, c] | 100 | 100 | 58 | 80 |
| LWC (mL m$^{-3}$)[b] | 0.190 | 0.258 | NA | NA |
| ALWC (µg m$^{-3}$)[b, d] | NA | NA | 69 | 515 |
| f$_{44}$[e] | 0.16 | 0.21 | 0.042 ± 0.006 | 0.097 ± 0.004 |
| f$_{60}$[e] | 0.007 | 0.004 | 0.016 ± 0.003 | 0.010 ± 0.001 |
| OM:OC[b, f] | 1.9 | 2.2 | 1.5 ± 0.1 | 1.9 ± 0.1 |
| O:C[b, f] | 0.58 | 0.8 | 0.24 ± 0.04 | 0.56 ± 0.03 |
| H:C[b, f] | 1.37 | 1.29 | 1.65 ± 0.03 | 1.60 ± 0.01 |
| OS$_C$[b, f] | -0.21 | 0.32 | -1.17 ± 0.08 | -0.48 ± 0.06 |

[a]Start collection times given in local time; [b]Average values corresponding to the collection times of individual samples; [c]Average RH was assumed to be 100% for fog samples, as super-saturation levels could not be measured; [d]ALWC is an average of E-AIM and ISORROPIA modeled data for the sampling period; [e]Fractional abundance of a mass fragment (f$_X$) was calculated as the ratio between that fragment signal and the total organic concentration; [f]Elemental ratios were calculated according to Aiken et al. (2008).



**Table 2:** Summary of FT-ICR MS formula assignment data. Mass, O:C, H:C, OS_C, DBE, AI, C_n, and O_n values represent mathematical averages based on formula assignment, with standard deviation provided. These values were obtained using equations 1-7.

| | | All | CHO | CHNO | CHOS | CHNOS |
|---|---|---|---|---|---|---|
| SPC0106F | Number | 2824 | 1158 (41%) | 744 (26%) | 619 (22%) | 303 (11%) |
| | Mass | 368.44 ± 94.21 | 359.05 ± 101.35 | 342.88 ± 90.69 | 404.68 ± 83.64 | 393.12 ± 61.72 |
| | O:C | 0.479 ± 0.16 | 0.415 ± 0.13 | 0.503 ± 0.14 | 0.488 ± 0.14 | 0.642 ± 0.18 |
| | H:C | 1.30 ± 0.36 | 1.21 ± 0.32 | 1.14 ± 0.27 | 1.56 ± 0.34 | 1.53 ± 0.34 |
| | $OS_C$ | -0.345 ± 0.45 | -0.379 ± 0.42 | -0.133 ± 0.33 | -0.583 ± 0.51 | -0.247 ± 0.43 |
| | DBE | 7.24 ± 3.65 | 8.29 ± 3.67 | 8.35 ± 3.01 | 4.93 ± 3.08 | 5.20 ± 2.93 |
| | $AI_{mod}$ | 0.24 ± 0.22 | 0.31 ± 0.20 | 0.32 ± 0.21 | 0.08 ± 0.12 | 0.06 ± 0.11 |
| | $C_N$ | 17.2 ± 5.2 | 18.3 ± 5.5 | 15.7 ± 4.9 | 17.8 ± 4.8 | 15 ± 3.7 |
| | $O_N$ | 7.8 ± 2.3 | 7.4 ± 2.5 | 7.6 ± 2.2 | 8.3 ± 2.0 | 9.1 ± 1.3 |
| SPC0201F | Number | 1671 | 890 (53%) | 427 (26%) | 212 (13%) | 142 (8%) |
| | Mass | 360.12 ± 97.52 | 358.18 ± 108.61 | 364.33 ± 90.94 | 360.12 ± 78.27 | 359.66 ± 63.56 |
| | O:C | 0.577 ± 0.18 | 0.509 ± 0.13 | 0.617 ± 0.14 | 0.592 ± 0.15 | 0.858 ± 0.24 |
| | H:C | 1.31 ± 0.35 | 1.18 ± 0.29 | 1.23 ± 0.26 | 1.74 ± 0.23 | 1.77 ± 0.20 |
| | $OS_C$ | -0.158 ± 0.43 | -0.161 ± 0.41 | 0.007 ± 0.30 | -0.551 ± 0.45 | -0.050 ± 0.50 |
| | DBE | 6.83 ± 3.53 | 8.05 ± 3.38 | 7.49 ± 2.69 | 3.01 ± 1.81 | 2.93 ± 1.13 |
| | $AI_{mod}$ | 0.22 ± 0.21 | 0.30 ± 0.19 | 0.22 ± 0.20 | 0.00 ± 0.03 | 0.00 ± 0.00 |
| | $C_N$ | 15.8 ± 5.0 | 16.9 ± 5.2 | 15.4 ± 4.4 | 14.4 ± 4.0 | 11.8 ± 3.4 |
| | $O_N$ | 8.7 ± 2.7 | 8.5 ± 3.0 | 9.2 ± 2.4 | 8.2 ± 2.0 | 9.5 ± 1.6 |
| BO0204N | Number | 1634 | 808 (49%) | 732 (45%) | 42 (3%) | 52 (3%) |
| | Mass | 364.99 ± 100.13 | 358.24 ± 105.13 | 373.63 ± 98.27 | 332.45 ± 54.27 | 374.39 ± 50.93 |
| | O:C | 0.433 ± 0.14 | 0.377 ± 0.11 | 0.480 ± 0.14 | 0.405 ± 0.11 | 0.652 ± 0.17 |
| | H:C | 1.13 ± 0.32 | 1.12 ± 0.30 | 1.04 ± 0.22 | 1.96 ± 0.10 | 1.77 ± 0.14 |
| | $OS_C$ | -0.262 ± 0.41 | -0.368 ± 0.38 | -0.080 ± 0.32 | -1.152 ± 0.26 | -0.462 ± 0.39 |
| | DBE | 9.26 ± 3.94 | 9.42 ± 4.03 | 9.99 ± 3.11 | 1.31 ± 0.64 | 3.08 ± 0.9 |
| | $AI_{mod}$ | 0.36 ± 0.20 | 0.38 ± 0.19 | 0.39 ± 0.18 | 0.00 ± 0.00 | 0.00 ± 0.00 |
| | $C_N$ | 18.0 ± 5.6 | 18.9 ± 5.7 | 17.6 ± 5.5 | 15.0 ± 3.5 | 13.9 ± 3.4 |
| | $O_N$ | 7.5 ± 2.4 | 7.0 ± 2.5 | 8.1 ± 2.3 | 5.8 ± 0.9 | 8.6 ± 0.7 |
| BO0213D | Number | 2753 | 1097 (40%) | 1123 (41%) | 249 (9%) | 284 (10%) |
| | Mass | 361.82 ± 96.19 | 351.26 ± 102.62 | 360.99 ± 94.43 | 354.82 ± 75.64 | 412.02 ± 76.3 |
| | O:C | 0.498 ± 0.19 | 0.424 ± 0.15 | 0.555 ± 0.18 | 0.435 ± 0.16 | 0.617 ± 0.21 |
| | H:C | 1.37 ± 0.37 | 1.25 ± 0.34 | 1.26 ± 0.27 | 1.9 ± 0.22 | 1.8 ± 0.18 |
| | $OS_C$ | -0.372 ± 0.5 | -0.399 ± 0.46 | -0.152 ± 0.39 | -1.027 ± 0.44 | -0.562 ± 0.45 |





| | | | | | |
|---|---|---|---|---|---|
| DBE | $6.64 \pm 3.65$ | $7.81 \pm 3.88$ | $7.44 \pm 2.6$ | $1.8 \pm 1.43$ | $3.19 \pm 1.47$ |
| $AI_{mod}$ | $0.21 \pm 0.21$ | $0.29 \pm 0.21$ | $0.22 \pm 0.20$ | $0.01 \pm 0.06$ | $0.00 \pm 0.00$ |
| $C_N$ | $16.7 \pm 5.3$ | $17.9 \pm 5.7$ | $15.8 \pm 4.9$ | $16 \pm 4.9$ | $16 \pm 4.7$ |
| $O_N$ | $7.8 \pm 2.5$ | $7.2 \pm 2.4$ | $8.3 \pm 2.5$ | $6.3 \pm 1.4$ | $9.1 \pm 1.6$ |



**Table 3:** Summary of the possible identified molecular formulas from the present study. Identical formulas from the literature are provided with their references. Additional possible identified molecular formulas are listed in Table S1.

| Formula | SPC0106F | SPC0201F | BO0204N | BO0213D | Possible Identity | Reference |
|---------|----------|----------|---------|---------|-------------------|-----------|
| $C_6H_4N_2O_5$ | X | X | X | X | 2,4-dinitrophenol | a |
| $C_6H_5NO_3$ | X | X | X | X | 4-nitrophenol | a, b |
| $C_6H_5NO_4$ | X | X | X | X | 4-nitrocatechol | b |
| $C_6H_6O_3$ | | | | X | phenol SOA (pyrogallol) | c-e |
| $C_6H_6O_4$ | | X | | X | phenol SOA | |
| $C_6H_6O_5$ | | | | X | phenol SOA | f |
| $C_6H_{10}O_5$ | | X | | X | levoglucosan | b-e, g, h |
| $C_7H_5NO_5$ | X | X | X | X | 3-nitrosalicylic acid | a |
| $C_7H_6O_2$ | X | X | | X | benzoic acid | b, h |
| $C_7H_6O_3$ | X | X | X | X | 4-hydroxybenzoic acid | c-f |
| $C_7H_7NO_3$ | X | X | X | X | 2-methyl-4-nitrophenol | a, b |
| $C_7H_7NO_4$ | X | X | X | X | 4-nitroguaiacol | a |
| $C_7H_8O_3$ | X | X | | X | guaiacol SOA | |
| $C_7H_8O_4$ | X | X | X | X | guaiacol SOA | |
| $C_7H_8O_5$ | X | X | X | X | guaiacol SOA | |
| $C_7H_8O_6$ | X | X | | X | guaiacol SOA | |
| $C_7H_{12}O_5$ | | X | X | X | 3-hydroxy-4,4-dimethylglutaric acid | i |
| $C_8H_6O_4$ | X | X | X | X | phthalic acid | b-e, h |
| $C_8H_8O_2$ | X | X | X | X | o-toluic acid | b, h |
| $C_8H_8O_3$ | X | X | X | X | vanillin | b-e |
| $C_8H_8O_4$ | X | X | X | X | vanillic acid | c-e, g, h |
| $C_8H_{10}O_3$ | X | X | | X | syringol | f-h, j |
| $C_8H_{10}O_4$ | X | X | | X | syringol SOA | |
| $C_8H_{10}O_5$ | X | X | X | X | syringol aqSOA | j |
| $C_8H_{10}O_6$ | X | X | X | X | syringol aqSOA | f |
| $C_8H_{10}O_7$ | | X | | X | syringol aqSOA | f |
| $C_8H_{12}O_6$ | X | X | X | X | 3-methyl-1,2,3-butanetricarboxylic acid | i |
| $C_9H_8O_3$ | X | X | X | X | coumaryc acid | b-e |
| $C_9H_{10}O_3$ | X | X | X | X | acetovanillone | c-e, f, j, k |
| $C_9H_{10}O_4$ | X | X | X | X | syringaldehyde | c-e, g, h, j |
| $C_9H_{10}O_5$ | X | X | X | X | syringic acid | c-e, g, h |
| $C_9H_{11}NO_3$ | X | X | X | X | tyrosine | |
| $C_9H_{12}O_3$ | X | X | | X | 4-methylsyringol | h |
| $C_9H_{14}O_3$ | X | X | | X | ketolimononaldehyde | l |
| $C_9H_{14}O_4$ | X | X | | X | pinic acid | i |
| $C_9H_{16}O_4$ | X | X | X | X | azelaic acid | c-e, h |





| | | | | | | |
|---|---|---|---|---|---|---|
| $C_{10}H_{12}O_2$ | X | X | | X | eugenol | h |
| $C_{10}H_{12}O_4$ | X | X | X | X | acetosyringone | c-e |
| $C_{10}H_{14}O_3$ | X | X | | X | ketopinic acid | i |
| $C_{10}H_{16}O_3$ | X | X | | X | pinonic acid | c-e, i |

*References are (a) Kitanovski et al. (2012); (b) Desyaterik et al. (2013); (c) Pietrogrande et al. (2014a); (d) Pietrogrande et al. (2014b); (e) Pietrogrande et al. (2015); (f) Yu et al. (2016); (g) Dzepina et al. (2015); (h) Mazzoleni et al. (2007); (i) He et al. (2014); (j) Yu et al. (2014); (k) Lin et al. (2015); and (l) Nguyen et al. (2013).



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
