# Peer review of "Molecular insights on aging and aqueous phase processing from ambient biomass burning emissions-influenced Po Valley fog and aerosol"

_Atmospheric Chemistry and Physics, 2018_

## Referee Comment (RC1) · Anonymous Referee #1 · 1 Jun 2018

**GENERAL COMMENTS**

This paper describes high resolution mass spectrometry (HRMS) and NMR analysis of water soluble organic compounds (WSOC) in four samples selected to represent fresh and aged aerosol particles as well as fresh and aged fog droplets. Results of this work generally support the importance of aqueous processing of organic aerosols. The data set is definitively interesting and worth publishing. The paper can be improved by addressing several issues described below. In addition, for the amount of new information presented in this paper it is too long. I would recommend shortening it and making it more focused on new findings.

[Figure]

My main criticism of this paper is its reliance on just one sample of each type (fresh aerosol, aged aerosol, fresh fog, and aged fog) to draw far reaching conclusions about chemical processes that are responsible for aging of WSOC. Furthermore, samples come from completely different dates making it quite difficult to faithfully compare them. This is much less satisfying and convincing than the approach taken by the authors in Gilardoni et al. (2016) that looked at the fog dissipation events. The authors rely on aerosol mass spectrometry (AMS) analysis, specifically on the f44-f60 correlation plots, to classify the samples. To prove to the readers that this classification works as expected the authors should compare at least two HRMS data sets from different filters that are supposed to be identical based on the f44-f60 AMS classification. Why not take a couple of samples (as opposed to a single sample) corresponding to closely spaced points in Figure 1 and compare their HRMS data? I am willing to bet that the authors would find very different molecular composition for these supposedly similar samples. If this is the case, the comparison of the HRMS data between different conditions becomes more difficult and potentially not even possible.

SPECIFIC COMMENTS

Abstract: To avoid confusing the readers I recommend removing the prefix "high resolution" in front of Tof-AMS in the abstract and in the text. I understand that this how this instrument was called when it was designed but it is a stretch to call it a high resolution instrument especially in a paper that relies on FT-ICR as the main method. Using simple "ToF-AMS" should be sufficient.

P1L21: particles containing organosulfates might activate more easily accounting for the higher fraction of organosulfates in fog droplets compared to aerosol particles.

P1L25: it would be useful to also add ranges for O:C and H:C for the "fresh" samples so that one can compare

P5L26: it should also be pointed out that valence of 3 is used for N, so the extra double bond in the nitrooxy compounds is not counted

P5L29: the authors should warn the readers that OS developed by Kroll et a. (2011) only works for CHO compounds, and that it also fails for peroxides. This formula cannot be used for CHOS and CHON compounds. The authors should check their text so that they do not over interpret results from this formula

P12L20: the fact that these compounds show in a single daylight sample does not constitute proof that these compounds are related to photochemical properties. This is one example of several statements made by the authors for which they do not have sufficient data. To claim something like this, they would need to demonstrate presence of these compounds in many daylight samples (not one!) and absence in many nighttime samples.

P13L15: another example where a conclusion is made (about sulfite radical involvement) without having needed data to prove it

Figure 5: What message is conveyed by this figure that cannot be more easily conveyed with average O:C values? I do not see how it helps interpret the data. Two versions of this figure exist, one in the text and one in the SI section. I would just keep it in the SI section or remove altogether.

Figure 7: it would help explain how the spectra were normalized before the subtraction. The result of the subtraction obviously would depend on the choices made in the normalization.

Table 3 and Table S1: it would be useful to specify peak abundance (such as very high, high, medium, minor or something similar). Also I would point out in the caption that the "identities" specified in one of the columns are for reference only – the fact that formulas match does not mean that this where these compounds came from.

S2: it states it there that the assignments were cut off above m/z 500 but assigned peaks in figure S3 go beyond 600

S2: please explain the "rule of 13" – this must be some sort of a mass spectrometry

jargon

EDITORIAL COMMENTS

P1L34: remove "evolving"

P2L7: "and more" -> "and other compounds"

P2L14: this sentence needs a revision (incompatible list items)

P6L26: "rich of" -> "containing"

P10L30: two sets of references need to be joined in one

P10L31: "act" -> "acted"

Figure 4: the choice of colors makes it hard to differentiate between them

Table 2: "mass" -> "molecular weight (g/mol)"

---

## Referee Comment (RC2) · Anonymous Referee #2 · 1 Jul 2018

This manuscript presents molecular-level analyses of fresh versus aged fog and samples influenced by biomass burning. The authors aim to explore the potential importance of aqueous phase processing on alteration of organic matter chemical compositions. The authors reported that aged aerosols and fresh fog samples show similarity in composition, indicating the possibility of aged aerosols that served as fog nuclei.

One of my major concerns for this manuscript is that the authors attributed the CHON and CHOS compounds exclusively to organonitrates and organosulfates based on FT–ICR MS analyses, but this is not supported by NMR spectra! This seems to be a major finding but it was not discussed in great detail. It looks to me that other types of organic
nitrogen and organosulfur compounds may contribute to formation of detected CHON and CHOS that need further investigations.

In addition, nitro groups (R-NO2) and nitrooxy groups (R-ONO2) are different. They have distinct formation processes and physiochemical properties as well (e.g., lifetime against hydrolysis). The authors need to be clear when discussing their findings in context of literature.

For the results and discussion, the current form of manuscript is a bit lengthy and repetitive when reporting the FT-ICR MS data. A more concise presentation will greatly improve the readers' reading experience.

Also, reactions in aerosol liquid water content and in fog should be discussed separately. Based on the results presented (with only 1 sample in each category), the aqueous processing does alter the chemical compositions of organic matter, but the pathways are rather inconclusive.

Overall, this is still a nice case study that provides useful information. Below I provide a few more specific comments for the authors' consideration and clarification.

Specific Comments:

1) Page 4, lines 7-9: The aerosol filter extracts were filtered with 0.45 $\mu$m PTFE membrane, while the fog water was filtered through 47 mm quartz fiber filters. What is the pore size of 47 mm quartz fiber filters? Why did the authors use two different filtering methods here? Since the FT–ICR MS analysis is very sensitive, potential artifacts (even trace amounts) during sample preparation should be avoided.

2) Page 7, line 6: Does "SOA-like" mean oxygenated/or functionalized/or fragmented? It is not clear here.

3) Page 10, lines 25-35: Since aged aerosols could act as fog nuclei, scavenging of organosulfates resided in aged aerosols into fog might have contributed to the observed organosulfates in fresh fog water. Based on the data presented, I don't really

see direct evidence here showing that aqueous processing leads to CHOS production. Similarly, on Page 13 lines 12-17: the authors concluded that the current data provide strong evidence of aqueous processing that dominates the production of S-containing organic matter. I would tone down this statement.

4) Page 12, line 32: "hygroscopic" is a better term to describe aged/oxygenated organics that contribute to droplet formation.

5) Page 13, lines 8-9: it is confusing when the authors stated "some evidence of dimerization" here. This was not presented in "results and discussion" but suddenly mentioned in summary.

---

## Author Comment (AC2) · 21 Aug 2018

Author responses to comments from RC2 are given below in blue font. The original referee comments are provided in *black italicized font.*

*This manuscript presents molecular-level analyses of fresh versus aged fog and samples influenced by biomass burning. The authors aim to explore the potential importance of aqueous phase processing on alteration of organic matter chemical compositions. The authors reported that aged aerosols and fresh fog samples show similarity in composition, indicating the possibility of aged aerosols that served as fog nuclei. One of my major concerns for this manuscript is that the authors attributed the CHON and CHOS compounds exclusively to organonitrates and organosulfates based on FT–ICR MS analyses, but this is not supported by NMR spectra! This seems to be a major finding but it was not discussed in great detail. It looks to me that other types of organic nitrogen and organosulfur compounds may contribute to formation of detected CHON and CHOS that need further investigations. In addition, nitro groups (R-NO2) and nitrooxy groups (R-ONO2) are different. They have distinct formation processes and physiochemical properties as well (e.g., lifetime against hydrolysis). The authors need to be clear when discussing their findings in context of literature.*

*For the results and discussion, the current form of manuscript is a bit lengthy and repetitive when reporting the FT-ICR MS data. A more concise presentation will greatly improve the readers' reading experience. Also, reactions in aerosol liquid water content and in fog should be discussed separately. Based on the results presented (with only 1 sample in each category), the aqueous processing does alter the chemical compositions of organic matter, but the pathways are rather inconclusive.*

*Overall, this is still a nice case study that provides useful information. Below I provide a few more specific comments for the authors' consideration and clarification.*

We thank the referee for their helpful comments. As suggested, we made several edits to improve the readability and reduce redundancies in the manuscript. We also clarified the comparisons of fresh fog to aged fog compositions and fresh aerosol to aged aerosol compositions (Sections 3.3.3 and 3.3.4 respectively). Another comparison between the two types of samples was made because of the interesting observation of similar compositions between the aged aerosol and the fresh fog. It is plausible that some reactions in aerosol liquid water may also occur during fog activation as droplets begin to grow, helping to explain the similar compositions between these two samples of different types.

The results of the $^1$H-NMR and FT-ICR MS measurements are only apparently in contradiction. While the $^1$H-NMR analyses were performed on bulk aqueous samples, the electrospray ionization FT-ICR MS analysis requires the removal of inorganic ions present in the bulk aqueous extracts. Reversed phase SPE cartridges were used to isolate the water-soluble organic aerosol components. However, some losses of low molecular weight and ionic water-soluble organic compounds are expected. This may have included the low-molecular weight alkyl amines observed by $^1$H-NMR analysis. Furthermore, the negative ion electrospray favors the detection of acidic compounds and thus is not ideal for the detection of reduced nitrogen or sulfur compounds. A statement was added to the end of section 2.4 (p. 6 line 19-22) to clarify this: "The resulting data set represents the SPE-recovered higher molecular weight water soluble organic aerosol and is expected to predominantly contain acidic compounds due to the negative ion ESI analytical bias. The observed molecular compositions represent the oxidized fraction of the atmospheric samples thus, useful insights can be made with these limitations in mind."

The concern as to whether the [1]H-NMR data actually support the hypothesis of the occurrence of organic nitrates and organic sulfates in these samples can be reassured by the clear signals in the spectral regions where aliphatic hydrogen atoms in alpha position to such functional groups are expected to occur (specifically between 4 and 5 ppm chemical shift, as described by Hsieh et al. (2014)). However, it is not as clear as to whether the [1]H-NMR analysis provides the same information on the relative abundance of organic nitrates and organic sulfates between samples as derived from the FT-ICR MS datasets, as the same spectral region can host many other possible functionalities (e.g., esters, peroxides, hydroxy-carboxylic acids). In summary, [1]H-NMR spectroscopy was not specific enough to trace the abundance of organic nitrates and sulfates in these samples in a useful manner for comparison with the FT-ICR MS data. The N and S containing molecular formulas observed with FT-ICR were attributed to organonitrates, organosulfates and nitrooxy-organosulfates, based on a previous study using negative mode electrospray ionization and MS/MS analysis for functional group determination of water-soluble atmospheric organic matter (LeClair et al., 2012), and the observed O:N and O:S ratios, which we clarified in the text.

*Specific Comments:*

*1) Page 4, lines 7-9: The aerosol filter extracts were filtered with 0.45 um PTFE membrane, while the fog water was filtered through 47 mm quartz fiber filters. What is the pore size of 47 mm quartz fiber filters? Why did the authors use two different filtering methods here? Since the FT–ICR MS analysis is very sensitive, potential artifacts (even trace amounts) during sample preparation should be avoided.*

These method differences result from the laboratory methods for the different analysis techniques. Aerosol filter extractions were performed at Michigan Tech (MTU) for FT-ICR analysis (Section 2.4) and at the Institute of Atmospheric Sciences and Climate (CNR-ISAC) for total carbon, [1]H-NMR analysis and HR-ToF-AMS analysis for fog samples (Section 2.1). Fog and aerosol samples prepared at MTU for FT-ICR analysis were prepared consistently with 25 mm quartz fiber filters, as described in section 2.4. We are not aware of a uniform pore size for quartz fiber filters, due to the nature of the material. Sample blanks were used to correct for artifacts that may have been introduced by the quartz fiber filter due to the sensitivity of FT-ICR MS.  Additional statements were added to section 2.1 (p. 4 line 18-25): "The aerosol filters were extracted with deionized ultra-pure water (Milli-Q) in an ultrasonic bath for 1 h. The water extract was filtered with a 0.45 μm PTFE membrane in order to remove suspended particles. Fog water was filtered through 47 mm quartz fiber filters within a few hours of collection and conductivity and pH measurements were taken ... Aliquots of both aerosol water extracts and fog water prepared in this way were used to determine the total organic carbon content … and water soluble organic carbon (WSOC) concentration, (Rinaldi et al., 2007) as well as for [1]H-NMR analysis and HR-ToF-AMS analysis of fog samples described below (HR-ToF-AMS data for aerosol samples was collected in real time)." and 2.4 (p. 5 line 22-24): "Fog samples were later re-filtered using a 25 mm quartz filter before SPE. A portion of the aerosol filter samples were extracted with ultrapure water using sonication and the extracts were then filtered using a 25 mm quartz filter to remove insoluble materials…" to clarify these differences in methods.

*2) Page 7, line 6: Does "SOA-like" mean oxygenated/or functionalized/or fragmented? It is not clear here.*

We have revised the text to more accurately convey that the fog compositions were more oxidized, and thus more similar to SOA than the aerosols, according to the simplified source attribution scheme of Fig. 1. The text was changed to reflect this in the abstract (p.1 line 25-27): "Fog compositions were more

oxidized and "SOA-like" than aerosols as indicated by their NMR measured acyl vs alkoxyl ratios and the observed molecular formula similarity between the aged aerosol and fresh fog, implying that fog nuclei must be somewhat aged." In the $^1$H-NMR discussion (p. 8 line 3-5): "So, according to the simple source-attribution scheme based on the major $^1$H-NMR functionalities presented here, the fog compositions were more oxidized and "SOA-like" than aerosols." And in the conclusions (p. 14 line 18-20): "Overall, the fog composition was generally more oxidized and "SOA-like" than the aerosol, where the fresh fog composition was similar to the aged aerosol composition in both the $^1$H-NMR analysis and the molecular formula trends."

*3) Page 10, lines 25-35: Since aged aerosols could act as fog nuclei, scavenging of organosulfates resided in aged aerosols into fog might have contributed to the observed organosulfates in fresh fog water. Based on the data presented, I don't really see direct evidence here showing that aqueous processing leads to CHOS production. Similarly, on Page 13 lines 12-17: the authors concluded that the current data provide strong evidence of aqueous processing that dominates the production of S-containing organic matter. I would tone down this statement.*

We have revised this paragraph to include statements on nucleation scavenging. The revised paragraph (p. 12 line 3-13) now reads: "The unique molecular formulas found in the fresh fog (SPC0106F) were mostly of the $O_{5-13}S$ and $NO_{7-12}S$ subclasses. Organosulfates are known products of aqueous secondary processes, (Darer et al., 2011; Ervens et al., 2011; McNeill, 2015; Schindelka et al., 2013) and nucleation scavenging from the preceding fog nuclei composition likely plays a significant role as well (Darer et al., 2011; Gilardoni et al., 2014; Herckes et al., 2007; Hu et al., 2011). The aromatic organosulfates and nitooxy-organosulfates observed in fresh biomass burning aerosol (Staudt et al., 2014) were not observed here. In general, organosulfates are the products of aqueous-phase SOA reactions which are expected to be enhanced at acidic pH (Ervens et al., 2011; McNeill et al., 2012; Noziere et al., 2010). Because the pH of SPC0106F was only slightly acidic at 5.81, we propose that the formation of these organosulfates may have been promoted by low LWC, and thus relatively high solute concentrations, during the activation of the fog droplets or possibly in the fully formed fog droplets. Organosulfates may also efficiently nucleate droplets, leading to their eventual presence in the fog samples."

The conclusion statement (now p. 14 line 21-23) was revised to the following: "CHOS and CHNOS formulas were detected with high frequencies in samples with high water content during collection (all samples except BO0204N). This supports an enhanced production of S-containing SOA species via reactions in the aqueous phase."

*4) Page 12, line 32: "hygroscopic" is a better term to describe aged/oxygenated organics that contribute to droplet formation.*

 This sentence (now p. 14 line 4-5) now reads: "Hygroscopic species are expected to enhance droplet formation, indicating that organics acting as fog nuclei must be somewhat aged."

*5) Page 13, lines 8-9: it is confusing when the authors stated "some evidence of dimerization" here. This was not presented in "results and discussion" but suddenly mentioned in summary.*

We have revised the text to remove all references to dimerization.

References:

Darer, A. I., Cole-Filipiak, N. C., O'Connor, A. E., and Elrod, M. J.: Formation and Stability of Atmospherically Relevant Isoprene-Derived Organosulfates and Organonitrates, Environ Sci Technol, 45, 1895-1902, 2011.

Ervens, B., Turpin, B. J., and Weber, R. J.: Secondary organic aerosol formation in cloud droplets and aqueous particles (aqSOA): a review of laboratory, field and model studies, Atmos Chem Phys, 11, 11069-11102, 2011.

Gilardoni, S., Massoli, P., Giulianelli, L., Rinaldi, M., Paglione, M., Pollini, F., Lanconelli, C., Poluzzi, V., Carbone, S., Hillamo, R., Russell, L. M., Facchini, M. C., and Fuzzi, S.: Fog scavenging of organic and inorganic aerosol in the Po Valley, Atmos Chem Phys, 14, 6967-6981, 2014.

Herckes, P., Chang, H., Lee, T., and Collett, J. L.: Air pollution processing by radiation fogs, Water Air Soil Poll, 181, 65-75, 10.1007/s11270-006-9276-x, 2007.

Hsieh, P. H., Xu, Y. M., Keire, D. A., and Liu, J.: Chemoenzymatic synthesis and structural characterization of 2-O-sulfated glucuronic acid-containing heparan sulfate hexasaccharides, Glycobiology, 24, 681-692, 10.1093/glycob/cwu032, 2014.

Hu, K. S., Darer, A. I., and Elrod, M. J.: Thermodynamics and kinetics of the hydrolysis of atmospherically relevant organonitrates and organosulfates, Atmos Chem Phys, 11, 8307-8320, 2011.

LeClair, J. P., Collett, J. L., and Mazzoleni, L. R.: Fragmentation Analysis of Water-Soluble Atmospheric Organic Matter Using Ultrahigh-Resolution FT-ICR Mass Spectrometry, Environ Sci Technol, 46, 4312-4322, 2012.

McNeill, V. F., Woo, J. L., Kim, D. D., Schwier, A. N., Wannell, N. J., Sumner, A. J., and Barakat, J. M.: Aqueous-Phase Secondary Organic Aerosol and Organosulfate Formation in Atmospheric Aerosols: A Modeling Study, Environ Sci Technol, 46, 8075-8081, 2012.

McNeill, V. F.: Aqueous Organic Chemistry in the Atmosphere: Sources and Chemical Processing of Organic Aerosols, Environ Sci Technol, 49, 1237-1244, 10.1021/es5043707, 2015.

Noziere, B., Ekstrom, S., Alsberg, T., and Holmstrom, S.: Radical-initiated formation of organosulfates and surfactants in atmospheric aerosols, Geophys Res Lett, 37, 2010.

Rinaldi, M., Emblico, L., Decesari, S., Fuzzi, S., Facchini, M. C., and Librando, V.: Chemical characterization and source apportionment of size-segregated aerosol collected at an urban site in sicily, Water Air Soil Poll, 185, 311-321, 2007.

Schindelka, J., Iinuma, Y., Hoffmann, D., and Herrmann, H.: Sulfate radical-initiated formation of isoprene-derived organosulfates in atmospheric aerosols, Faraday Discuss, 165, 237-259, 2013.

Staudt, S., Kundu, S., Lehmler, H. J., He, X. R., Cui, T. Q., Lin, Y. H., Kristensen, K., Glasius, M., Zhang, X. L., Weber, R. J., Surratt, J. D., and Stone, E. A.: Aromatic organosulfates in atmospheric aerosols: Synthesis, characterization, and abundance, Atmos Environ, 94, 366-373, 2014.

---

## Author Response (AR1)

Aggregated Author Response Documents with Tracked Manuscript Changes

All *Referee comments are included below in italic black font* and the Author responses are in

blue font.

Summary of Manuscript Revisions: In addition to specifically addressing each of the reviewer comments, we removed some of the less important technical description of the FT-ICR MS composition in sections 3.3 and 3.4. The manuscript has been edited for grammar corrections and clarity.
* * *
Author responses to comments from RC1 are given below in blue font. The original referee comments are provided in *black italicized font.*

*This paper describes high resolution mass spectrometry (HRMS) and NMR analysis of water soluble organic compounds (WSOC) in four samples selected to represent fresh and aged aerosol particles as well as fresh and aged fog droplets. Results of this work generally support the importance of aqueous processing of organic aerosols. The data set is definitively interesting and worth publishing. The paper can be improved by addressing several issues described below. In addition, for the amount of new information presented in this paper it is too long. I would recommend shortening it and making it more focused on new findings.*

*My main criticism of this paper is its reliance on just one sample of each type (fresh aerosol, aged aerosol, fresh fog, and aged fog) to draw far reaching conclusions about chemical processes that are responsible for aging of WSOC. Furthermore, samples come from completely different dates making it quite difficult to faithfully compare them. This is much less satisfying and convincing than the approach taken by the authors in Gilardoni et al. (2016) that looked at the fog dissipation events. The authors rely on aerosol mass spectrometry (AMS) analysis, specifically on the f44-f60 correlation plots, to classify the samples. To prove to the readers that this classification works as expected the authors should compare at least two HRMS data sets from different filters that are supposed to be identical based on the f44-f60 AMS classification. Why not take a couple of samples (as opposed to a single sample) corresponding to closely spaced points in Figure 1 and compare their HRMS data? I am willing to bet that the authors would find very different molecular composition for these supposedly similar samples. If this is the case, the comparison of the HRMS data between different conditions becomes more difficult and potentially not even possible.*

We thank the referee for their appreciation of the manuscript content and helpful comments. We understand the main criticism raised by the referee about the limited number of analyzed samples, in fact we are working on the analysis of a larger database collected during a more recent field experiment, specifically designed to investigate aqueous phase processing. Nevertheless, the complexity of ultrahigh resolution FT-ICR MS database generated from a single aerosol or fog water sample set some limitations on the number of samples that could be analyzed within a reasonable amount of time. One of the goals of this study is to analyze extremely different samples of aerosol and fog water in term of ageing of organic content and impact of wood burning emissions, to fully deploy the potential of ultrahigh resolution MS and identify the subset of information relevant for a larger database analysis. In addition, the low time resolution of ultrahigh resolution MS and $^1$H-NMR analysis (hours or days) compared to HR-ToF-AMS analysis (minutes) requires a different approach in the data analysis compared to what was done in

Gilardoni et al. (2016), where we were able to follow the formation and dissipation of single fog events. For this reason, we rely on the high time resolution of HR-ToF-AMS analysis to identify aerosol samples for further analysis with ultrahigh resolution MS and [1]H-NMR techniques as described in the text.

The selection of aerosol samples was based on a detailed characterization of the field experiment data reported in a previous publication (Gilardoni et al., 2016), and is not the subject of the present manuscript. Instead, for the selection of fog water samples, we used the approach commonly employed by HR-ToF-AMS users for organic aerosol, investigating the $f_{44}$ and $f_{60}$ space, since it is recognized that $f_{44}$ is a marker of ageing organic content and $f_{60}$ is a proxy for wood burning organic molecules (Cubison et al., 2011). We are aware that this representation is an oversimplification of the complexity of organic fog water content, thus this approach is here employed exclusively to spot marked differences in term of different sources and atmospheric history of organic content. The following was added to section 3.1 on p. 7, line 5-8 to clarify this: "The $f_{44}$ vs. $f_{60}$ space was previously proposed to represent biomass burning vs. atmospheric aerosol aging (Cubison et al., 2011) and was extended here to fog samples. This representation is an oversimplification of the complexity of organic molecules in fog water, employed here exclusively to note the major differences in terms of emission sources and atmospheric history."

As mentioned above the goal of this work was to study very different samples. Even with these selections, we still observe many of the same molecular formulas across samples as indicated in the van Krevelen diagram Fig. 5. Thus, it is unlikely that similar samples as defined in Fig. 1 would yield "*very different molecular composition*". However, the day to day composition of aerosol and fog and its evolution with respect to the local meteorology is the focus of a future publication.

Overall, this study provides evidence of the potential of combining high-field spectroscopic techniques (Hertkorn et al., 2007) to trace chemical changes in ambient aerosol in specific environmental conditions. We performed a screening of the possible organic compositions using HR-ToF-AMS and a simple functional group analysis by [1]H-NMR, and clearly chose extreme conditions in the chemical space (Fig. 1) for further in-depth chemical analyses. This is progressive with respect to previous explorative approaches employing combined [1]H-NMR and FT-ICR MS methods for aerosol analysis (Schmitt-Kopplin et al., 2010) which provided little information on the actual environmental conditions affecting the composition of the aerosol.

*SPECIFIC COMMENTS*

*Abstract: To avoid confusing the readers I recommend removing the prefix "high resolution" in front of Tof-AMS in the abstract and in the text. I understand that this how this instrument was called when it was designed but it is a stretch to call it a high resolution instrument especially in a paper that relies on FT-ICR as the main method. Using simple "ToF-AMS" should be sufficient.*

We agree that the terms high resolution and ultrahigh resolution are similar. However, the term "high resolution" refers specifically to the subclass of mass spectrometers that use the time of flight (ToF) to derive the measured mass. There are ToF instruments without high resolution. For this reason, we consistently refer to the instruments as HR-ToF-AMS and ultrahigh resolution FT-ICR MS (or simply FT-ICR MS).

*P1L21: particles containing organosulfates might activate more easily accounting for the higher fraction of organosulfates in fog droplets compared to aerosol particles.*

This sentence (now on p. 1 line 18-19) was changed to reflect the intended observational nature of the statement: "Higher numbers of organonitrates were observed in aerosol, and higher numbers of organosulfates were observed in fog water." While we agree that organosulfate compounds in aerosol may be more hygroscopic and aid in droplet activation, it is well documented that organosulfate compounds form in the aqueous phase of cloud/fog and wet aerosol particles (Darer et al., 2011; Ervens et al., 2011; Schindelka et al., 2013; Herrmann et al., 2015; McNeill, 2015); therefore, pre-existing organosulfates in aerosol may only indicate multiple cycles of fog formation and evaporation. We have added a statement about organosulfates in the fog from nucleation scavenging on p. 12 line 5-6: "…and nucleation scavenging from the preceding fog nuclei composition likely plays a significant role as well…"

*P1L25: it would be useful to also add ranges for O:C and H:C for the "fresh" samples so that one can compare*

This sentence (now p.1 line 21-23) was edited to compare the values of both the "fresh" and "aged" samples in O:C and H:C values, and the actual range of these values was added to compare more easily: "The average O:C and H:C values from FT-ICR MS were higher in the samples with an "aged" influence (O:C = 0.50-0.58 and H:C = 1.31-1.37) compared to those with "fresh" influence (O:C = 0.43-0.48 and H:C = 1.13-1.30)."

*P5L26: it should also be pointed out that valence of 3 is used for N, so the extra double bond in the nitrooxy compounds is not counted*

A sentence (p. 6 line 8-9) was added to clarify that the calculated DBE values do not include double bonds formed by pentavalent nitrogen, and tetravalent or hexavalent sulfur: "Note that S and O are divalent in equation (3); additional unsaturated bonds associated with pentavalent nitrogen, and tetravalent or hexavalent sulfur are not included in this DBE calculation."

*P5L29: the authors should warn the readers that OS developed by Kroll et al. (2011) only works for CHO compounds, and that it also fails for peroxides. This formula cannot be used for CHOS and CHON compounds. The authors should check their text so that they do not over interpret results from this formula*

We thank the referee for this reminder. We modified the $OS_C$ calculation as described in Kroll et al. (2011) to more accurately calculate $OS_C$ for formulas containing nitrogen and sulfur. However, this requires the assumption that when N is present it represents a nitrate functional group and when S is present it represents a sulfate functional group. This assumption is reasonable considering we analyzed the samples using negative ion electrospray, however, it is still an assumption. Furthermore, we assume that unstable peroxide species would not survive sample storage and sample preparation for analysis by FT ICR. This modified calculation is now included in the text (p. 6 line 9-12) along with the necessary assumption: "The average oxidation state of carbon ($OS_C$) in the molecular formulas was estimated using equation (4), based on the approximation described in Kroll et al. (2011); note that the inclusion of nitrogen and sulfur affects the oxidation state of carbon, and equation (4) assumes both are fully oxidized."

*P12L20: the fact that these compounds show in a single daylight sample does not constitute proof that these compounds are related to photochemical properties. This is one example of several statements made by the authors for which they do not have sufficient data. To claim something like this, they would*

*need to demonstrate presence of these compounds in many daylight samples (not one!) and absence in many nighttime samples.*

In our discussion of the results, we suggest that these molecular formulas may have been formed from photolysis reactions, because of the ~8.6 hours of daylight in sample collection. Furthermore, this was the major difference between the two aerosol samples. Thus, it would not have been appropriate to ignore such a major difference. We agree that photochemistry is not directly responsible for the $N_2$ and $N_3$ formulas, as there is a trend between the presence of these formulas and $NO_X$ concentration during sample collection for all samples. We have modified this statement (now p. 13 line 27-28) describing them as such: "Compared to the other samples, BO0213D was collected during relatively high $NO_X$ conditions, as well as high humidity and aerosol liquid water content compared to the other aerosol sample."

*P13L15: another example where a conclusion is made (about sulfite radical involvement) without having needed data to prove it*

Here we related our observation of organosulfates in these samples to literature sources as a suggestion for their origin. We agree that the involvement of sulfite radical is unlikely since most of the samples were collected at night. This statement (now p. 14 line 21-23) was revised to the following: "CHOS and CHNOS formulas were detected with high frequencies in samples with high water content during collection (all samples except BO0204N). This provided some evidence of the production of S-containing SOA species by reactions in the aqueous phase."

*Figure 5: What message is conveyed by this figure that cannot be more easily conveyed with average O:C values? I do not see how it helps interpret the data. Two versions of this figure exist, one in the text and one in the SI section. I would just keep it in the SI section or remove altogether.*

The van Krevelen space is useful for visualizing both oxidation (O:C) and saturation (H:C). In this work, we have identified thousands of individual molecular formulas. We observed both highly oxidized species and highly saturated species in the same samples, and this plot is able to show these differences. Furthermore, some of these formulas are observed in all of the samples, and some are unique to the individual samples. The van Krevelen plots in Fig. S2 includes all of these formulas, where the symbols in Fig. 5 are differentiated to indicate unique or common molecular formulas. The unique formulas in each sample help to further illustrate the differences between samples (outlined in multiple sections), hence we included two versions of the plot.

*Figure 7: it would help explain how the spectra were normalized before the subtraction. The result of the subtraction obviously would depend on the choices made in the normalization.*

The normalization procedure and its importance has been expanded upon in the supplemental text (p. S2): "The total ion abundance of the identified monoisotopic molecular formulas reported for each sample was determined by their summation. Then, these values were used to normalize the individual ion abundances within each sample using a ratio of the individual ion intensity to this total ion abundance. Then, the values were rescaled using a normalization constant (10,000). This normalization procedure was done to remove analytical biases introduced by trace contaminants with high electrospray efficiency."

*Table 3 and Table S1: it would be useful to specify peak abundance (such as very high, high, medium, minor or something similar). Also, I would point out in the caption that the "identities" specified in one of the columns are for reference only – the fact that formulas match does not mean that this where these compounds came from.*

We thank the reviewer for this helpful insight. Table 3 was moved into Table S1. We modified the extended Table S1 to indicate the normalized abundance of each formula. We revised the caption for Table S1 to include additional clarification regarding the nature of molecular formulas vs. chemical structures: "Table S1: Summary of the literature structural insights associated with the identified molecular formulas observed in this study. Because the identified molecular formulas may represent a variety of structural isomers, we note that matched molecular formulas do not necessarily correspond to the same molecular structure or atmospheric origin. The normalized abundances are indicated for each sample, where "ND" (not detected), "Low" (≤ 3%), "Med", (> 3% and ≤ 15%), "High" (> 15% and ≤ 50%) and "Very High" (> 50%). Molecular formulas from the literature are provided with their references." An additional paragraph was added to section 2.4 (p. 6 line 22-24) to further clarify the difference between molecular formulas and chemical structures: "Furthermore, it is important to note that the individual molecular formulas likely represent a mixture of structural isomers co-existing in atmospheric organic matter, as recently observed for deep-sea organic matter (Zark et al., 2017)." We also removed all references to specific IUPAC style chemical compound names, e.g. "2,4-dinitrophenol" was changed to "dinitrophenol" in the discussion of formulas matched to the previous literature.

*S2: it states it there that the assignments were cut off above m/z 500 but assigned peaks in figure S3 go beyond 600*

We describe the *de novo* cutoff in greater detail in the supplemental text (p. S2), as it does not perform a hard cut off for formulas of $m/z > 500$: "A de novo cut-off at $m/z$ 500 was applied, indicating that no new formula assignments would occur above $m/z$ 500, unless the formula was part of an existing $CH_2$ homologous series that began at a point lower than $m/z$ 500. This is necessary because the number of possible molecular formulas increases at higher values."

*S2: please explain the "rule of 13" – this must be some sort of a mass spectrometry jargon*

Descriptions of the rule of 13 and the nitrogen rule, were added to the supplemental text (p. S2): "The rule of 13 checks for a reasonable number of heteroatoms in a formula. A base formula ($C_nH_{n+r}$) can be generated for any measured mass by solving: $\frac{M}{13} = n + \frac{r}{13}$ (Pavia, 2009). Then, the maximum number of "large atoms" (C, O, N, S) in a formula is defined as the mass divided by 13, because substituting for a heteroatom (O, N or S) involves a substitution for at least one carbon. This maximum number is then compared to the actual number of "large atoms" in a formula, and those formulas exceeding the maximum number are rejected. The nitrogen rule removes formulas with odd masses that do not contain an odd number of nitrogen atoms, and even masses that do not contain an even number (or no) nitrogen atoms; this is due to the odd numbered valence of nitrogen (Pavia, 2009)."

*EDITORIAL COMMENTS*

*P1L34: remove "evolving"*

This sentence (now p. 1 line 32-p. 2 line 1) now reads: "Atmospheric organic aerosol particles are comprised of a complex mixture of numerous individual organic compounds, produced by direct

emissions and secondary processes, of which a significant impact is from transformations in the aqueous phase."

*P2L7: "and more" -> "and other compounds"*

This sentence (now p. 2 line 9-11) now reads: "Biomass burning products include simple organic acids, sugars and anhydrosugars, substituted phenols, polycyclic aromatic hydrocarbons, and other compounds, depending on the type of fuel and burn conditions…"

*P2L14: this sentence needs a revision (incompatible list items)*

This sentence (now p. 2 line 17-19) now reads: "Atmospheric chemistry models are currently unable to replicate several key aspects of SOA, including SOA concentration levels, chemical oxidation states, degree of functionalization, and the occurrence of high molecular weight compounds, such as atmospheric humic-like substances."

*P6L26: "rich of" -> "containing"*

This sentence (now p. 7 line 18-20) was changed to use the same consistent descriptor with all portions describing the 3 categories of interest, and now reads: "These categories are: SOA (enriched in acyl groups, H-C-C=O), biomass burning aerosol (enriched in alkoxyls, H-C-O, and aromatics), and marine organic aerosol (enriched in aliphatic groups other than acyls and alkoxyls, mainly amines and sulfoxy groups)."

*P10L30: two sets of references need to be joined in one*

This paragraph was revised to be easier to understand. The respective set of references have been combined into the new revised paragraph. The paragraph is provided in response to the next comment below.

*P10L31: "act" -> "acted"*

This entire paragraph was revised for clarity. The revised paragraph (p. 12 line 3-13) now reads: "The unique molecular formulas found in the fresh fog (SPC0106F) were mostly of the $O_{5-13}S$ and $NO_{7-12}S$ subclasses. Organosulfates are known products of aqueous secondary processes, (Darer et al., 2011; Ervens et al., 2011; McNeill, 2015; Schindelka et al., 2013) and nucleation scavenging from the preceding fog nuclei composition likely plays a significant role as well (Darer et al., 2011; Gilardoni et al., 2014; Herckes et al., 2007; Hu et al., 2011). The aromatic organosulfates and nitooxy-organosulfates observed in fresh biomass burning aerosol (Staudt et al., 2014) were not observed here. In general, organosulfates are the products of aqueous-phase SOA reactions which are expected to be enhanced at acidic pH (Ervens et al., 2011; McNeill et al., 2012; Noziere et al., 2010). Because the pH of SPC0106F was only slightly acidic at 5.81, we propose that the formation of these organosulfates may have been promoted by low LWC, and thus relatively high solute concentrations, during the activation of the fog droplets or possibly in the fully formed fog droplets. Organosulfates may also efficiently nucleate droplets, leading to their eventual presence in the fog samples."

*Figure 4: the choice of colors makes it hard to differentiate between them*

This Figure was revised from a yellow scale, to a more discerning color palette including yellow, blue and red.

*Table 2: "mass" -> "molecular weight (g/mol)"*

The term "Mass" was changed in the Table to "Molecular weight (Da)" to include the unit. Though the Da unit is numerically equivalent to g mol$^{-1}$, Da is used more frequently as a unit in mass spectrometry literature.

References:

[revised manuscript text omitted]

Author responses to comments from RC2 are given below in blue font. The original referee comments are provided in *black italicized font.*

*This manuscript presents molecular-level analyses of fresh versus aged fog and samples influenced by biomass burning. The authors aim to explore the potential importance of aqueous phase processing on alteration of organic matter chemical compositions. The authors reported that aged aerosols and fresh fog samples show similarity in composition, indicating the possibility of aged aerosols that served as fog nuclei. One of my major concerns for this manuscript is that the authors attributed the CHON and CHOS compounds exclusively to organonitrates and organosulfates based on FT–ICR MS analyses, but this is not supported by NMR spectra! This seems to be a major finding but it was not discussed in great detail. It looks to me that other types of organic nitrogen and organosulfur compounds may contribute to formation of detected CHON and CHOS that need further investigations. In addition, nitro groups (R-NO2) and nitrooxy groups (R-ONO2) are different. They have distinct formation processes and physiochemical properties as well (e.g., lifetime against hydrolysis). The authors need to be clear when discussing their findings in context of literature.*

*For the results and discussion, the current form of manuscript is a bit lengthy and repetitive when reporting the FT-ICR MS data. A more concise presentation will greatly improve the readers' reading experience. Also, reactions in aerosol liquid water content and in fog should be discussed separately. Based on the results presented (with only 1 sample in each category), the aqueous processing does alter the chemical compositions of organic matter, but the pathways are rather inconclusive.*

*Overall, this is still a nice case study that provides useful information. Below I provide a few more specific comments for the authors' consideration and clarification.*

We thank the referee for their helpful comments. As suggested, we made several edits to improve the readability and reduce redundancies in the manuscript. We also clarified the comparisons of fresh fog to aged fog compositions and fresh aerosol to aged aerosol compositions (Sections 3.3.3 and 3.3.4 respectively). Another comparison between the two types of samples was made because of the interesting observation of similar compositions between the aged aerosol and the fresh fog. It is plausible that some reactions in aerosol liquid water may also occur during fog activation as droplets begin to grow, helping to explain the similar compositions between these two samples of different types.

The results of the $^1$H-NMR and FT-ICR MS measurements are only apparently in contradiction. While the $^1$H-NMR analyses were performed on bulk aqueous samples, the electrospray ionization FT-ICR MS analysis requires the removal of inorganic ions present in the bulk aqueous extracts. Reversed phase SPE cartridges were used to isolate the water-soluble organic aerosol components. However, some losses of low molecular weight and ionic water-soluble organic compounds are expected. This may have included the low-molecular weight alkyl amines observed by $^1$H-NMR analysis. Furthermore, the negative ion electrospray favors the detection of acidic compounds and thus is not ideal for the detection of reduced nitrogen or sulfur compounds. A statement was added to the end of section 2.4 (p. 6 line 19-22) to clarify this: "The resulting data set represents the SPE-recovered higher molecular weight water soluble organic aerosol and is expected to predominantly contain acidic compounds due to the negative ion ESI analytical bias. The observed molecular compositions represent the oxidized fraction of the atmospheric samples thus, useful insights can be made with these limitations in mind."

The concern as to whether the [1]H-NMR data actually support the hypothesis of the occurrence of organic nitrates and organic sulfates in these samples can be reassured by the clear signals in the spectral regions where aliphatic hydrogen atoms in alpha position to such functional groups are expected to occur (specifically between 4 and 5 ppm chemical shift, as described by Hsieh et al. (2014)). However, it is not as clear as to whether the [1]H-NMR analysis provides the same information on the relative abundance of organic nitrates and organic sulfates between samples as derived from the FT-ICR MS datasets, as the same spectral region can host many other possible functionalities (e.g., esters, peroxides, hydroxy-carboxylic acids). In summary, [1]H-NMR spectroscopy was not specific enough to trace the abundance of organic nitrates and sulfates in these samples in a useful manner for comparison with the FT-ICR MS data. The N and S containing molecular formulas observed with FT-ICR were attributed to organonitrates, organosulfates and nitrooxy-organosulfates, based on a previous study using negative mode electrospray ionization and MS/MS analysis for functional group determination of water-soluble atmospheric organic matter (LeClair et al., 2012), and the observed O:N and O:S ratios, which we clarified in the text.

*Specific Comments:*

*1) Page 4, lines 7-9: The aerosol filter extracts were filtered with 0.45 um PTFE membrane, while the fog water was filtered through 47 mm quartz fiber filters. What is the pore size of 47 mm quartz fiber filters? Why did the authors use two different filtering methods here? Since the FT–ICR MS analysis is very sensitive, potential artifacts (even trace amounts) during sample preparation should be avoided.*

These method differences result from the laboratory methods for the different analysis techniques. Aerosol filter extractions were performed at Michigan Tech (MTU) for FT-ICR analysis (Section 2.4) and at the Institute of Atmospheric Sciences and Climate (CNR-ISAC) for total carbon, [1]H-NMR analysis and HR-ToF-AMS analysis for fog samples (Section 2.1). Fog and aerosol samples prepared at MTU for FT-ICR analysis were prepared consistently with 25 mm quartz fiber filters, as described in section 2.4. We are not aware of a uniform pore size for quartz fiber filters, due to the nature of the material. Sample blanks were used to correct for artifacts that may have been introduced by the quartz fiber filter due to the sensitivity of FT-ICR MS.  Additional statements were added to section 2.1 (p. 4 line 18-25): "The aerosol filters were extracted with deionized ultra-pure water (Milli-Q) in an ultrasonic bath for 1 h. The water extract was filtered with a 0.45 µm PTFE membrane in order to remove suspended particles. Fog water was filtered through 47 mm quartz fiber filters within a few hours of collection and conductivity and pH measurements were taken ... Aliquots of both aerosol water extracts and fog water prepared in this way were used to determine the total organic carbon content … and water soluble organic carbon (WSOC) concentration, (Rinaldi et al., 2007) as well as for [1]H-NMR analysis and HR-ToF-AMS analysis of fog samples described below (HR-ToF-AMS data for aerosol samples was collected in real time)." and 2.4 (p. 5 line 22-24): "Fog samples were later re-filtered using a 25 mm quartz filter before SPE. A portion of the aerosol filter samples were extracted with ultrapure water using sonication and the extracts were then filtered using a 25 mm quartz filter to remove insoluble materials…" to clarify these differences in methods.

*2) Page 7, line 6: Does "SOA-like" mean oxygenated/or functionalized/or fragmented? It is not clear here.*

We have revised the text to more accurately convey that the fog compositions were more oxidized, and thus more similar to SOA than the aerosols, according to the simplified source attribution scheme of Fig. 1. The text was changed to reflect this in the abstract (p.1 line 25-27): "Fog compositions were more

oxidized and "SOA-like" than aerosols as indicated by their NMR measured acyl vs alkoxyl ratios and the observed molecular formula similarity between the aged aerosol and fresh fog, implying that fog nuclei must be somewhat aged." In the [1]H-NMR discussion (p. 8 line 3-5): "So, according to the simple source-attribution scheme based on the major [1]H-NMR functionalities presented here, the fog compositions were more oxidized and "SOA-like" than aerosols." And in the conclusions (p. 14 line 18-20): "Overall, the fog composition was generally more oxidized and "SOA-like" than the aerosol, where the fresh fog composition was similar to the aged aerosol composition in both the [1]H-NMR analysis and the molecular formula trends."

*3) Page 10, lines 25-35: Since aged aerosols could act as fog nuclei, scavenging of organosulfates resided in aged aerosols into fog might have contributed to the observed organosulfates in fresh fog water. Based on the data presented, I don't really see direct evidence here showing that aqueous processing leads to CHOS production. Similarly, on Page 13 lines 12-17: the authors concluded that the current data provide strong evidence of aqueous processing that dominates the production of S-containing organic matter. I would tone down this statement.*

We have revised this paragraph to include statements on nucleation scavenging. The revised paragraph (p. 12 line 3-13) now reads: "The unique molecular formulas found in the fresh fog (SPC0106F) were mostly of the $O_{5-13}S$ and $NO_{7-12}S$ subclasses. Organosulfates are known products of aqueous secondary processes, (Darer et al., 2011; Ervens et al., 2011; McNeill, 2015; Schindelka et al., 2013) and nucleation scavenging from the preceding fog nuclei composition likely plays a significant role as well (Darer et al., 2011; Gilardoni et al., 2014; Herckes et al., 2007; Hu et al., 2011). The aromatic organosulfates and nitooxy-organosulfates observed in fresh biomass burning aerosol (Staudt et al., 2014) were not observed here. In general, organosulfates are the products of aqueous-phase SOA reactions which are expected to be enhanced at acidic pH (Ervens et al., 2011; McNeill et al., 2012; Noziere et al., 2010). Because the pH of SPC0106F was only slightly acidic at 5.81, we propose that the formation of these organosulfates may have been promoted by low LWC, and thus relatively high solute concentrations, during the activation of the fog droplets or possibly in the fully formed fog droplets. Organosulfates may also efficiently nucleate droplets, leading to their eventual presence in the fog samples."

The conclusion statement (now p. 14 line 21-23) was revised to the following: "CHOS and CHNOS formulas were detected with high frequencies in samples with high water content during collection (all samples except BO0204N). This supports an enhanced production of S-containing SOA species via reactions in the aqueous phase."

*4) Page 12, line 32: "hygroscopic" is a better term to describe aged/oxygenated organics that contribute to droplet formation.*

 This sentence (now p. 14 line 4-5) now reads: "Hygroscopic species are expected to enhance droplet formation, indicating that organics acting as fog nuclei must be somewhat aged."

*5) Page 13, lines 8-9: it is confusing when the authors stated "some evidence of dimerization" here. This was not presented in "results and discussion" but suddenly mentioned in summary.*

We have revised the text to remove all references to dimerization.

[revised manuscript text omitted]

Formatted Table ... [134]
Formatted ... [135]
Formatted ... [136]
Formatted ... [138]
Formatted ... [137]
Formatted ... [139]
Formatted ... [140]
Formatted ... [141]
Formatted ... [142]
Formatted ... [143]
Formatted ... [144]
Formatted ... [145]
Formatted ... [147]
Formatted ... [146]
Formatted ... [148]
Formatted ... [149]
Formatted ... [150]
Formatted ... [151]
Formatted ... [152]
Formatted ... [153]
Formatted ... [154]
Formatted ... [156]
Formatted ... [155]
Formatted ... [157]
Formatted ... [158]
Formatted ... [159]
Formatted ... [160]
Formatted ... [161]
Formatted ... [162]
Formatted ... [163]
Formatted ... [164]
Formatted ... [165]
Formatted ... [166]
Formatted ... [167]
Formatted ... [168]
Formatted ... [169]
Formatted ... [170]
Formatted ... [171]
Formatted ... [172]
Formatted ... [173]
Formatted ... [174]
Formatted ... [176]
Formatted ... [175]
Formatted ... [177]
Formatted ... [178]
Formatted ... [179]

[revised manuscript text omitted]

English (US)

| **Page 2: [1] Formatted** | mabrege | **8/20/18 3:30:00 PM** |

English (US)

| **Page 2: [1] Formatted** | mabrege | **8/20/18 3:30:00 PM** |

English (US)

| **Page 2: [1] Formatted** | mabrege | **8/20/18 3:30:00 PM** |

English (US)

| **Page 2: [1] Formatted** | mabrege | **8/20/18 3:30:00 PM** |

English (US)

| **Page 2: [2] Formatted** | mabrege | **8/20/18 3:30:00 PM** |

English (US)

| **Page 2: [2] Formatted** | mabrege | **8/20/18 3:30:00 PM** |

English (US)

| **Page 2: [2] Formatted** | mabrege | **8/20/18 3:30:00 PM** |

English (US)

| **Page 2: [3] Formatted** | mabrege | **8/20/18 3:30:00 PM** |

English (US)

| **Page 2: [3] Formatted** | mabrege | **8/20/18 3:30:00 PM** |

English (US)

| **Page 2: [3] Formatted** | mabrege | **8/20/18 3:30:00 PM** |

English (US)

| **Page 2: [3] Formatted** | mabrege | **8/20/18 3:30:00 PM** |

English (US)

| **Page 2: [3] Formatted** | mabrege | **8/20/18 3:30:00 PM** |

English (US)

| **Page 2: [3] Formatted** | mabrege | **8/20/18 3:30:00 PM** |

English (US)

| **Page 2: [4] Formatted** | mabrege | **8/20/18 3:30:00 PM** |

English (US)

| **Page 2: [4] Formatted** | mabrege | **8/20/18 3:30:00 PM** |

English (US)

**Page 2: [4] Formatted**        mabrege        **8/20/18 3:30:00 PM**
English (US)

**Page 2: [5] Formatted**        mabrege        **8/20/18 3:30:00 PM**
English (US)

**Page 2: [5] Formatted**        mabrege        **8/20/18 3:30:00 PM**
English (US)

**Page 2: [5] Formatted**        mabrege        **8/20/18 3:30:00 PM**
English (US)

**Page 2: [5] Formatted**        mabrege        **8/20/18 3:30:00 PM**
English (US)

**Page 2: [6] Formatted**        mabrege        **8/20/18 3:30:00 PM**
English (US)

**Page 2: [6] Formatted**        mabrege        **8/20/18 3:30:00 PM**
English (US)

**Page 2: [7] Formatted**        mabrege        **8/20/18 3:30:00 PM**
English (US)

**Page 2: [7] Formatted**        mabrege        **8/20/18 3:30:00 PM**
English (US)

**Page 2: [7] Formatted**        mabrege        **8/20/18 3:30:00 PM**
English (US)

**Page 2: [7] Formatted**        mabrege        **8/20/18 3:30:00 PM**
English (US)

**Page 2: [7] Formatted**        mabrege        **8/20/18 3:30:00 PM**
English (US)

**Page 2: [7] Formatted**        mabrege        **8/20/18 3:30:00 PM**
English (US)

**Page 2: [7] Formatted**        mabrege        **8/20/18 3:30:00 PM**
English (US)

**Page 2: [7] Formatted**        mabrege        **8/20/18 3:30:00 PM**
English (US)

| **Page 2: [7] Formatted** | **mabrege** | **8/20/18 3:30:00 PM** |

English (US)

| **Page 2: [7] Formatted** | **mabrege** | **8/20/18 3:30:00 PM** |

English (US)

| **Page 2: [7] Formatted** | **mabrege** | **8/20/18 3:30:00 PM** |

English (US)

| **Page 2: [7] Formatted** | **mabrege** | **8/20/18 3:30:00 PM** |

English (US)

| **Page 2: [7] Formatted** | **mabrege** | **8/20/18 3:30:00 PM** |

English (US)

| **Page 2: [7] Formatted** | **mabrege** | **8/20/18 3:30:00 PM** |

English (US)

| **Page 3: [8] Formatted** | **mabrege** | **8/20/18 3:30:00 PM** |

English (US)

| **Page 3: [9] Formatted** | **mabrege** | **8/20/18 3:30:00 PM** |

English (US)

| **Page 3: [10] Formatted** | **mabrege** | **8/20/18 3:30:00 PM** |

English (US)

| **Page 3: [11] Formatted** | **mabrege** | **8/20/18 3:30:00 PM** |

English (US)

| **Page 3: [12] Formatted** | **mabrege** | **8/20/18 3:30:00 PM** |

English (US)

| **Page 3: [13] Formatted** | **mabrege** | **8/20/18 3:30:00 PM** |

English (US)

[revised manuscript text omitted]

*Correspondence to:* Lynn R. Mazzoleni (lrmazzol@mtu.edu)

**Table of Contents**

**1. FT-ICR MS data processing and molecular formula assignment review**

The individual transient scans of FT-ICR MS data for each sample were reviewed manually and the unacceptable scans with an abrupt change in the total ion current were removed; the remaining transient scans were co-added together to create the working file for each sample (this helped to increase signal to noise and enhance sensitivity). Molecular formula assignments were made as previously described (Mazzoleni et al., 2010; Putman et al., 2012; Zhao et al., 2013; Dzepina et al., 2015) using Sierra Analytics Composer software (version 1.0.5) within the limits of: $C_{2-200}H_{4-1000}O_{1-20}N_{0-3}S_{0-1}$. Masses were calculated from measured $m/z$ values, assuming an ion charge of -1 from the electrospray. The calculator uses a $CH_2$ Kendrick mass defect (KMD) analysis to sort homologous ion series and extend the molecular formula assignments to higher masses (Hughey et al., 2001; Kujawinski and Behn, 2006). A de novo cut-off at $m/z$ 500 was applied, indicating that no new formula assignments would occur above $m/z$ 500, unless the formula was part of an existing $CH_2$ homologous series that began at a point lower than $m/z$ 500. This is necessary because the number of possible molecular formulas increases at higher values. The minimum relative abundance required for molecular formula assignment was > 10 times the estimated signal-to-noise ratio, determined for each sample between $m/z$ 900–1000. Only integer values up to 40 were allowed for the double bond equivalents (DBE). The data set was manually reviewed to remove: formulas with an absolute error > 3 ppm, elemental ratios that were not chemically sensible (such as O:C > 3 or H:C < 0.3), and formulas which violated the rule of 13 or violated the nitrogen rule. The rule of 13 checks for a reasonable number of heteroatoms in a formula. A base formula ($C_nH_{n+r}$) can be generated for any measured mass by solving: $\frac{M}{13} = n + \frac{r}{13}$ (Pavia, 2009). Then, the maximum number of "large atoms" (C, O, N, S) in a formula is defined as the mass divided by 13, because substituting for a heteroatom (O, N or S) involves a substitution for at least one carbon. This maximum number is then compared to the actual number of "large atoms" in a formula, and those formulas exceeding the maximum number are rejected. The nitrogen rule removes formulas with odd masses that do not contain an odd number of nitrogen atoms, and even masses that do not contain an even number (or no) nitrogen atoms; this is due to the odd numbered valence of nitrogen (Pavia, 2009). Molecular formulas that contained $^{13}C$ or $^{34}S$ were also removed from the data set. Homologous series with large gaps in the DBE trend were removed, as well as homologous series with a length of one. The assigned formulas were also analyzed with consideration to the DBE and oxygen number trends, (Herzsprung et al., 2014) where unreliable formula assignments were also removed.

**2. Ultrahigh resolution FT-ICR MS results**

The total ion abundance of the identified monoisotopic molecular formulas reported for each sample was determined by their summation. Then, these values were used to normalize the individual ion abundances within each sample using a ratio of the individual ion intensity to this total ion abundance. Then, the values were rescaled using a normalization constant (10,000). This normalization procedure was done to remove analytical biases introduced by trace contaminants with high electrospray efficiency.

Reconstructed difference mass spectra of the assigned molecular formulas for both fog and aerosol samples are shown in Fig. S3. These difference mass spectra permit a direct comparison of the samples using normalized

relative abundances. The individual relative abundances were normalized by the total abundance of the assigned molecular formulas identified in each of the samples. In Fig. S3, the individual masses with higher abundances in either the positive or negative direction were substantially greater in one of the two samples, whereas the masses of similar abundance tended to cancel each other. To enhance the interpretation of the compositional differences, the individual masses were color-coded to represent the number of oxygen atoms in the assigned formula. Overall, we observed higher numbers of oxygen in the masses of the two samples with aged biomass burning emissions influence compared to the two samples with fresh biomass burning emissions influence. The molecular formulas assigned to the fresh samples had approximately 0-5 oxygen atoms over the mass range of 50-250 Da, 5-10 oxygen atoms over 250-550 Da, and a few molecular formulas were assigned with 10-15 oxygen atoms over 500-600 Da. In contrast, the aged samples had a large number of molecular formulas with 10-15 oxygen atoms in the range of 400-550 Da. This clearly shows a greater amount of oxidation in the aged influenced samples compared to the fresh influenced samples.

KMD diagrams can be used as useful tools to visualize the relationships between the many molecular formulas of complex mixtures such as atmospheric samples. We used Kendrick mass defect to sort the molecular formulas into $CH_2$ homologous series of identical heteroatom content and DBE, where the formulas in the same series differ only by a number of $CH_2$ units (Stenson et al., 2003). It should be noted that the presence of multiple formulas in the same homologous series does not necessarily imply a related chemical structure. The homologous series are visible as horizontal rows of formulas in Figs. S6 and S7. There were multiple homologous series per subclass, where the base formula for each series differ in DBE and increase in KMD to form an ensemble of "steps" within each subclass. In our samples individual CHO and CHNO subclasses had approximately 5-16 different homologous series, while CHOS and CHNOS subclasses had approximately 3-10 different homologous series. The number of homologous series in a subclass increased with oxygen number, and peaked near the median oxygen number, then decreased again towards the maximum number of oxygen; this led to fewer molecular formulas in subclasses with higher and lower oxygen numbers, and more formulas in subclasses near the median oxygen number. The subclasses with the highest numbers of molecular formulas per elemental group were: $O_7$, $NO_8$, $O_7S$ and $NO_9S$. It was atypical for the unique formulas of a sample to be completely unrelated to other formulas across the data set; often the unique formulas were extensions of homologous series that appeared across samples.

**3. FT-ICR MS data set**

An abbreviated list of the complete FT-ICR MS dataset is provided and is available on Digital Commons: http://digitalcommons.mtu.edu/chemistry-fp/98/

**4. Supplemental Figures and tables**

[Figure]

**Figure S1:** Distributions of the molecular formulas within all 64 elemental group subclasses for CHO, CHNO, CHOS and CHNOS groups as indicated in the Figure. The total number of molecular formulas for each SPE-recovered WSOC sample were split into two groups of unique and non-unique formulas; the darker shade represents formulas unique to a sample, (denoted in the Figure legend with an asterisk after the sample name, e.g. "Fresh Fog*") while the lighter shade represents common formulas. The sample names Fresh Fog, Aged Fog, Fresh Aerosol, and Aged Aerosol correspond to SPC0106F, SPC0201F, BO0204N, and BO0213D, respectively.

[Figure]

**Figure S2:** van Krevelen diagrams for the SPE-recovered WSOC by elemental group (rows) and sample (columns) as indicated in the Figure. Dashed lines represent H:C = 1.2 (horizontal), O:C = 0.6 (vertical) and $OS_C = 0$ (diagonal) as described in Tu et al. (2016). Formulas are color scaled to the number of oxygen atoms in the assigned formula. The sample names Fresh Fog, Aged Fog, Fresh Aerosol, and Aged Aerosol correspond to SPC0106F, SPC0201F, BO0204N, and BO0213D, respectively.

[Figure]

**Figure S3:** Reconstructed difference mass spectra for theoretical masses of assigned molecular formulas in the Po Valley samples with normalized relative abundance. Fresh influenced samples (SPC0106F and BO0204N) are plotted with positive abundance and aged influenced samples (SPC0201F and BO0213D) are plotted with negative abundance. Molecular compositions in both samples with the same mass and similar normalized relative abundance are reduced toward zero. The peaks in the mass spectra are color scaled to the number of oxygen atoms in the assigned molecular formula, where it can be observed that the aged samples shift towards species with higher oxygen numbers at lower masses, compared to the fresh samples. The sample names Fresh Fog, Aged Fog, Fresh Aerosol, and Aged Aerosol correspond to SPC0106F, SPC0201F, BO0204N, and BO0213D, respectively.

[Figure]

**Figure S4:** Oxygen difference trends for aerosol (a) and fog (b) samples. Abundance trends were calculated as in Figure 6 of the main text, and then the respective aged sample normalized abundance was subtracted from the fresh sample normalized abundance for each oxygen number value. A positive difference of abundance indicates an enhanced abundance of formulas in the fresh sample compared to the aged sample. Similarly, a negative difference of abundance indicates an enhanced abundance of formulas in the aged sample compared to the fresh sample. The sample names Fresh Fog, Aged Fog, Fresh Aerosol, and Aged Aerosol correspond to SPC0106F, SPC0201F, BO0204N, and BO0213D, respectively.

[Figure]

**Figure S5:** Double bond equivalent difference trends for aerosol (a) and fog (b) samples. Abundance trends were calculated as in Figure 6 of the main text, and then the respective aged sample normalized abundance was subtracted from the fresh sample normalized abundance for each integer double bond equivalent value. A positive difference of abundance indicates an enhanced abundance of formulas in the fresh sample compared to the aged sample. Similarly, a negative difference of abundance indicates an enhanced abundance of formulas in the aged sample compared to the fresh sample. The sample names Fresh Fog, Aged Fog, Fresh Aerosol, and Aged Aerosol correspond to SPC0106F, SPC0201F, BO0204N, and BO0213D, respectively.

[Figure]

**Figure S6:** Kendrick mass defect diagrams for each of the Po Valley samples, partitioned by elemental group (rows) and sample (columns) as indicated in the Figure. The molecular formulas unique to each sample are color scaled to the number of oxygen atoms in the assigned formula; grey points represent formulas which are common. Homologous series of molecular formulas are visible as horizontal rows of points, where formulas which are unique to a sample may make up all or only part of an individual homologous series. The sample names Fresh Fog, Aged Fog, Fresh Aerosol, and Aged Aerosol correspond to SPC0106F, SPC0201F, BO0204N, and BO0213D, respectively.

[Figure]

**Figure S7:** Kendrick mass defect diagrams for each of the Po Valley samples, partitioned by elemental group (rows) and sample (columns) as indicated in the Figure. Molecular formulas are color scaled to the number of oxygen atoms in the assigned formula. The sample names Fresh Fog, Aged Fog, Fresh Aerosol, and Aged Aerosol correspond to SPC0106F, SPC0201F, BO0204N, and BO0213D, respectively.

**Table S1:** Summary of the literature structural insights associated with the identified molecular formulas observed in this study. Because the identified molecular formulas may represent a variety of structural isomers, we note that matched molecular formulas do not necessarily correspond to the same molecular structure or atmospheric origin. The normalized abundances are indicated for each sample, where "ND" (not detected), "Low" (≤ 3%), "Med", (> 3% and ≤ 15%), "High" (> 15% and ≤ 50%) and "Very High" (> 50%). Molecular formulas from the literature are provided with their references.

| Formula | SPC0106F | SPC0201F | BO0204N | BO0213D | Possible Identity | Reference* |
|---|---|---|---|---|---|---|
| $C_4H_6O_5$ | ND | ND | ND | Low | syringol aqSOA | A |
| $C_5H_6O_3$ | ND | ND | ND | Low | ambient cloud water | B |
| $C_5H_6O_4$ | ND | Low | ND | Low | syringol aqSOA | A, B |
| $C_5H_6O_5$ | ND | ND | ND | Low | syringol aqSOA (ketoglutaric acid) | A, C-E |
| $C_5H_8O_4$ | ND | Med | ND | Low | ambient cloud water (methylsuccinic acid and glutatric acid) | B-F |
| $C_5H_8O_5$ | ND | ND | ND | Low | ambient cloud water (hydroxyglutaric acid) | B, G |
| $C_6H_4N_2O_5$ | High | Med | Low | Low | dinitrophenol | H |
| $C_6H_5NO_3$ | High | High | Med | Med | nitrophenol | B, H |
| $C_6H_5NO_4$ | High | High | Very High | Med | nitrocatechol | B, H |
| $C_6H_5NO_5$ | ND | ND | Med | Low | ambient cloud water | B |
| $C_6H_6O_3$ | ND | ND | ND | Low | pyrogallol | C-E |
| $C_6H_6O_4$ | ND | Low | ND | Med | phenol SOA | |
| $C_6H_6O_5$ | ND | ND | ND | Low | syringol aqSOA | A |
| $C_6H_8O_4$ | Low | Med | ND | Med | ambient cloud water | B |
| $C_6H_8O_6$ | ND | ND | ND | Low | syringol aqSOA | A |
| $C_6H_{10}O_3$ | ND | Low | ND | Low | ambient cloud water | B |
| $C_6H_{10}O_4$ | ND | Med | ND | Low | ambient cloud water (methylglutaric acid and adipic acid) | B-F |
| $C_6H_{10}O_5$ | ND | Low | ND | Low | levoglucosan | B-F, I |
| $C_6H_{10}O_6$ | ND | ND | ND | Low | dimethyltartaric acid | J |
| $C_7H_5NO_5$ | High | Med | Med | Med | nitrosalicylic acid | H |
| $C_7H_6O_2$ | Med | Med | ND | Low | ambient cloud water (benzoic acid) | B, F |
| $C_7H_6O_3$ | Med | Med | Med | Med | phenol aqSOA (dihydroxybenzaldehyde and hydroxybenzoic acid) | A, C-E |
| $C_7H_6O_4$ | Med | Low | Low | Med | phenol aqSOA | A |

| Formula | SPC0106F | SPC0201F | BO0204N | BO0213D | Possible Identity | Reference* |
|---|---|---|---|---|---|---|
| $C_7H_6O_5$ | ND | Low | ND | Low | syringol aqSOA | K |
| $C_7H_6N_2O_5$ | Med | Med | Med | Low | ambient cloud water | B, L |
| $C_7H_6N_2O_6$ | Med | ND | Med | Low | ambient cloud water | B, L |
| $C_7H_7NO_3$ | High | High | Med | High | methyl-nitrophenol | B, H |
| $C_7H_7NO_4$ | High | Med | Med | Med | ambient cloud water (nitroguaiacol and methyl-nitrocatechol) | B,H |
| $C_7H_7NO_5$ | Med | Low | Med | Med | ambient cloud water | B |
| $C_7H_8O_3$ | Med | Low | ND | High | guaiacol SOA | |
| $C_7H_8O_4$ | Med | Med | Low | Low | guaiacol SOA | |
| $C_7H_8O_5$ | Low | Med | Low | Med | guaiacol SOA | |
| $C_7H_8O_6$ | Low | Low | ND | Med | guaiacol SOA | |
| $C_7H_{10}O_4$ | Low | Med | Low | Med | ambient cloud water | L |
| $C_7H_{10}O_6$ | Low | Low | Low | Med | guaiacol aqSOA | A |
| $C_7H_{12}O_4$ | ND | Med | ND | Low | ambient cloud water (pimelic acid) | C-E, L |
| $C_7H_{12}O_5$ | ND | Low | Low | Med | biogenic SOA (hydroxy-dimethylglutaric acid) | G |
| $C_7H_{12}O_6S$ | ND | Low | ND | Low | ambient cloud water | L |
| $C_7H_{12}O_7$ | Low | ND | ND | ND | syringol aqSOA | A |
| $C_7H_{14}O_5S$ | Low | Med | ND | Low | dodecane aqSOA | L |
| $C_7H_{14}O_6S$ | ND | Low | ND | Low | ambient cloud water | L |
| $C_8H_5NO_4$ | Med | Low | Med | Low | ambient cloud water | B |
| $C_8H_6O_3$ | Med | Med | Low | Low | ambient cloud water and guaiacol aqSOA | A, B |
| $C_8H_6O_4$ | Med | Very High | Low | Low | ambient cloud water (phthalic acid) | B-F |
| $C_8H_6O_5$ | Med | Med | ND | Low | phenol aqSOA | A |
| $C_8H_7NO_3$ | Med | ND | Med | High | ambient cloud water | B |
| $C_8H_7NO_4$ | Med | Med | Med | High | ambient cloud water | B |
| $C_8H_7NO_5$ | High | Med | Med | High | ambient cloud water | B |
| $C_8H_8O_2$ | Med | Med | Low | Low | ambient cloud water (o-toluic acid) | B, F |
| $C_8H_8O_3$ | High | Med | Med | Low | ambient cloud water (vanillin) | B-E |

| Formula | SPC0106F | SPC0201F | BO0204N | BO0213D | Possible Identity | Reference* |
|---|---|---|---|---|---|---|
| $C_8H_8O_4$ | Med | Med | Low | Low | vanillic acid | C-F, I |
| $C_8H_9NO_3$ | Med | Med | Med | Low | ambient cloud water | B |
| $C_8H_9NO_4$ | High | Med | Very High | High | ambient cloud water | B |
| $C_8H_9NO_5$ | High | Med | High | High | ambient cloud water | B |
| $C_8H_{10}O_3$ | Med | Low | ND | Low | syringol | A,F, I, K |
| $C_8H_{10}O_4$ | Low | Med | ND | Low | syringol SOA | |
| $C_8H_{10}O_5$ | Med | Med | Low | Med | syringol aqSOA | K-L |
| $C_8H_{10}O_6$ | Low | Med | Low | Med | syringol aqSOA | A |
| $C_8H_{10}O_7$ | ND | Low | ND | Low | syringol aqSOA | A |
| $C_8H_{12}O_6$ | Med | Med | Low | Med | biogenic SOA (methyl-butanetricarboxylic acid) | G |
| $C_8H_{12}O_7S$ | ND | Low | ND | ND | ambient cloud water | L |
| $C_8H_{12}O_8S$ | Med | Low | ND | Low | ambient cloud water | L |
| $C_8H_{14}O_6S$ | Low | Med | ND | Low | ambient cloud water | L |
| $C_8H_{14}O_7$ | Med | ND | ND | ND | methylglyceric acid dimer | J |
| $C_8H_{14}O_7S$ | ND | Low | ND | ND | ambient cloud water | L |
| $C_8H_{16}O_6S$ | Med | Med | Low | Med | ambient cloud water | M |
| $C_9H_7NO_4$ | Med | Low | Med | Low | ambient cloud water | B |
| $C_9H_8O_2$ | Med | Low | Low | ND | ambient cloud water | B |
| $C_9H_8O_3$ | Med | Med | Med | Med | ambient cloud water (coumaryc acid) | B-E |
| $C_9H_8N_2O_6$ | Low | ND | Low | Med | ambient cloud water | L |
| $C_9H_9NO_3$ | Med | ND | Low | Med | ambient cloud water | B |
| $C_9H_9NO_3$ | Med | ND | Low | Med | laboratory brown carbon aqSOA | N |
| $C_9H_9NO_4$ | Med | Low | Med | Med | ambient cloud water | B |
| $C_9H_{10}O_3$ | Med | Low | Low | Low | laboratory brown carbon aqSOA (acetovanillone and dimethoxybenzaldehyde) | A, C-F, K, O |
| $C_9H_{10}O_4$ | Med | Med | Low | Low | syringol aqSOA (syringaldehyde) | C-F, I, K |
| $C_9H_{10}O_5$ | Med | Med | Low | Low | syringic acid | C-F, I |
| $C_9H_{11}NO_3$ | Med | Low | Low | Med | tyrosine | |

| Formula | SPC0106F | SPC0201F | BO0204N | BO0213D | Possible Identity | Reference* |
|---|---|---|---|---|---|---|
| $C_9H_{11}NO_4$ | Med | Low | Med | Med | ambient cloud water | B |
| $C_9H_{12}O_3$ | Low | Low | ND | Low | methylsyringol | F |
| $C_9H_{12}O_5$ | Med | Med | Low | Low | ambient cloud water | L |
| $C_9H_{12}O_7S$ | Low | ND | ND | ND | ambient cloud water | L |
| $C_9H_{12}N_2O_3$ | ND | ND | ND | Low | laboratory brown carbon aqSOA | O |
| $C_9H_{12}N_2O_3$ | ND | ND | ND | Low | laboratory brown carbon aqSOA | N |
| $C_9H_{13}NO_5$ | Low | ND | ND | Med | laboratory brown carbon aqSOA | N |
| $C_9H_{14}O_3$ | Low | Low | ND | Low | ketolimononaldehyde | P |
| $C_9H_{14}O_4$ | Med | Med | ND | Low | biogenic SOA (pinic acid) | G |
| $C_9H_{14}O_6S$ | Med | ND | ND | ND | ambient cloud water | L |
| $C_9H_{14}O_7S$ | Low | Med | ND | Low | ambient cloud water | L |
| $C_9H_{14}O_8S$ | Med | Med | ND | Low | ambient cloud water | L |
| $C_9H_{14}O_9S$ | Low | Low | ND | ND | ambient cloud water | L |
| $C_9H_{15}NO_8S$ | Very High | Very High | Low | Very High | ambient cloud water | L |
| $C_9H_{16}O_4$ | High | Med | Low | Low | azelaic acid | C-F |
| $C_9H_{16}O_6S$ | Med | Med | ND | Med | ambient cloud water | L |
| $C_9H_{16}O_7S$ | Low | Med | ND | Low | ambient cloud water | L |
| $C_9H_{16}O_8S$ | ND | Low | ND | ND | ambient cloud water | L |
| $C_9H_{18}O_6S$ | Med | Med | Low | Med | ambient cloud water, marine SOA | I, M |
| $C_9H_{18}O_8S$ | ND | ND | ND | Low | ambient cloud water | M |
| $C_{10}H_8O_3$ | Med | Low | Med | Low | syringol aqSOA | A |
| $C_{10}H_{10}O_4$ | High | Med | Med | ND | ferulic acid | C-E |
| $C_{10}H_{12}O_2$ | Low | Low | ND | Low | eugenol | F |
| $C_{10}H_{12}O_4$ | Med | Low | Low | Low | acetosyringone | C-E |
| $C_{10}H_{14}O_3$ | Low | Low | ND | Low | ketopinic acid | G |
| $C_{10}H_{14}O_5$ | Med | Med | Low | Low | ambient cloud water | L |
| $C_{10}H_{14}O_6$ | Med | Med | ND | Low | ambient cloud water | L |
| $C_{10}H_{14}O_7S$ | Med | ND | ND | ND | ambient cloud water | L |
| $C_{10}H_{14}O_8S$ | Low | Low | ND | ND | ambient cloud water | L |
| $C_{10}H_{16}O_3$ | Low | Low | ND | Low | pinonic acid | C-E, G |

| Formula | SPC0106F | SPC0201F | BO0204N | BO0213D | Possible Identity | Reference* |
|---|---|---|---|---|---|---|
| $C_{10}H_{16}O_6S$ | Med | ND | ND | ND | ambient cloud water | L |
| $C_{10}H_{16}O_7S$ | Med | High | ND | Low | ambient cloud water | L |
| $C_{10}H_{16}O_8S$ | ND | Med | ND | Low | ambient cloud water | L |
| $C_{10}H_{16}O_9S$ | Low | Low | ND | ND | ambient cloud water | L |
| $C_{10}H_{17}NO_7S$ | Very High | Very High | High | Very High | ambient cloud water | L |
| $C_{10}H_{17}NO_{10}S$ | Med | Med | Low | Med | ambient cloud water | L |
| $C_{10}H_{18}O_5S$ | ND | Med | ND | ND | ambient cloud water | L |
| $C_{10}H_{18}O_7S$ | Med | Med | ND | Low | ambient cloud water | L |
| $C_{10}H_{18}O_8S$ | Low | Med | ND | ND | ambient cloud water | L |
| $C_{10}H_{19}NO_9S$ | High | Med | Low | Med | ambient cloud water | M |
| $C_{10}H_{20}O_5S$ | Med | Med | Low | High | ambient cloud water | M |
| $C_{10}H_{20}O_6S$ | Med | Med | ND | Med | marine SOA | I |
| $C_{10}H_{20}O_7S$ | Low | Low | ND | Low | ambient cloud water | L |
| $C_{11}H_{10}O_8$ | ND | Low | ND | ND | phenol aqSOA | A |
| $C_{11}H_{21}NO_9S$ | Med | Med | Low | Med | ambient cloud water | M |
| $C_{11}H_{22}O_5S$ | Med | Med | Low | High | ambient cloud water | M |
| $C_{11}H_{22}O_6S$ | Med | Med | ND | Med | marine SOA | I |
| $C_{12}H_{10}O_2$ | Low | ND | Low | ND | phenol aqSOA | A, K |
| $C_{12}H_{10}O_3$ | Low | ND | Low | Low | phenol aqSOA | A |
| $C_{12}H_{10}O_4$ | ND | ND | Low | Low | phenol aqSOA | A |
| $C_{12}H_{10}O_7$ | Med | Med | Low | Low | syringol aqSOA | K |
| $C_{12}H_{10}N_2O_8$ | ND | ND | Low | ND | ambient cloud water | B |
| $C_{12}H_{11}NO_4$ | Med | ND | Low | Low | laboratory brown carbon aqSOA | N |
| $C_{12}H_{12}O_6$ | Med | Med | Low | Low | syringol aqSOA | K, L |
| $C_{12}H_{12}O_7$ | Med | Med | Low | Low | syringol aqSOA | A, K, L |
| $C_{12}H_{14}O_4$ | Med | Low | Low | Med | syringol aqSOA | A |
| $C_{12}H_{14}N_2O_4$ | ND | ND | ND | Low | laboratory brown carbon aqSOA | O |
| $C_{12}H_{14}N_2O_4$ | ND | ND | ND | Low | laboratory brown carbon aqSOA | N |
| $C_{12}H_{16}N_2O_5$ | ND | ND | ND | Low | laboratory brown carbon aqSOA | O |
| $C_{12}H_{17}NO_7$ | Low | Low | Low | Low | laboratory brown carbon aqSOA | N |

| Formula | SPC0106F | SPC0201F | BO0204N | BO0213D | Possible Identity | Reference* |
|---------|----------|----------|---------|---------|-------------------|------------|
| $C_{12}H_{20}O_7S$ | Med | Med | ND | Med | ambient cloud water | L |
| $C_{12}H_{22}O_7S$ | Med | Med | Low | Med | ambient cloud water | M |
| $C_{12}H_{23}NO_9S$ | Med | Med | Low | Med | ambient cloud water | M |
| $C_{12}H_{24}O_5S$ | Med | Low | Low | High | ambient cloud water | M |
| $C_{12}H_{24}O_6S$ | Med | Med | ND | Med | marine SOA | I |
| $C_{12}H_{26}O_4S$ | High | ND | Med | Low | ambient cloud water | M |
| $C_{13}H_{10}O_3$ | ND | ND | ND | Low | guaiacol aqSOA | A |
| $C_{13}H_{10}O_4$ | Med | Low | Low | ND | guaiacol aqSOA | A |
| $C_{13}H_{10}O_5$ | Med | ND | Low | Low | guaiacol aqSOA | A |
| $C_{13}H_{12}O_4$ | ND | ND | Low | Low | guaiacol aqSOA | A |
| $C_{13}H_{12}O_6$ | Med | Med | Low | Low | guaiacol aqSOA | A, L |
| $C_{13}H_{14}O_5$ | Med | Med | Med | Med | syringol aqSOA | A |
| $C_{13}H_{14}O_7$ | Med | Med | Low | Low | syringol aqSOA | A |
| $C_{13}H_{16}O_8$ | Med | ND | Low | ND | syringol aqSOA | A |
| $C_{13}H_{24}O_7S$ | Med | Med | Low | Med | ambient cloud water | M |
| $C_{13}H_{26}O_6S$ | Med | Low | Low | Med | ambient cloud water, marine SOA | I, M |
| $C_{14}H_{10}O_5$ | Med | Low | Low | Low | phenol aqSOA | A |
| $C_{14}H_{12}O_6$ | Med | Low | Low | Low | guaiacol aqSOA | A, L |
| $C_{14}H_{12}O_7$ | Med | Med | ND | Low | syringol aqSOA | A |
| $C_{14}H_{14}O_4$ | ND | ND | Med | Low | guaiacol aqSOA | A, I, K |
| $C_{14}H_{14}O_5$ | Med | Low | Low | Med | guaiacol aqSOA | A, K, L |
| $C_{14}H_{14}O_6$ | Med | Med | Low | Med | syringol aqSOA and guaiacol aqSOA | A, K |
| $C_{14}H_{14}O_8$ | Med | Med | ND | Low | Syringol aqSOA | A |
| $C_{14}H_{16}O_8$ | Med | Med | Low | Low | syringol aqSOA | A |
| $C_{14}H_{16}O_9$ | Med | Low | ND | Low | syringol aqSOA | A |
| $C_{14}H_{16}O_{10}$ | Low | Low | ND | Low | syringol aqSOA | A |
| $C_{14}H_{20}O_9$ | Low | Low | ND | Med | ambient cloud water | L |
| $C_{14}H_{24}O_8$ | ND | Low | ND | ND | ambient cloud water | L |
| $C_{14}H_{27}NO_9S$ | Med | Low | Low | Med | ambient cloud water | M |
| $C_{14}H_{28}O_5S$ | Med | Low | Low | Med | ambient cloud water | M |
| $C_{14}H_{28}O_6S$ | Med | Low | Low | Med | ambient cloud water | M |
| $C_{14}H_{30}O_4S$ | Med | ND | Low | Med | ambient cloud water | M |
| $C_{15}H_{14}O_6$ | Med | Low | Low | Low | syringol aqSOA | K, L |
| $C_{15}H_{14}O_8$ | Med | Med | Low | Low | syringol aqSOA | K |

| Formula | SPC0106F | SPC0201F | BO0204N | BO0213D | Possible Identity | Reference* |
|---|---|---|---|---|---|---|
| $C_{15}H_{16}O_6$ | Med | Med | Low | Low | syringol aqSOA | A, K, L |
| $C_{15}H_{16}O_8$ | Med | Med | ND | Low | syringol aqSOA | A |
| $C_{15}H_{16}O_9$ | Med | Med | ND | Low | syringol aqSOA | A, K, L |
| $C_{15}H_{18}O_7$ | Med | Med | Low | Low | syringol aqSOA | A, K, L |
| $C_{15}H_{18}O_9$ | Med | Med | Low | Low | syringol aqSOA | A |
| $C_{15}H_{18}O_{10}$ | Low | Low | ND | Low | syringol aqSOA | A |
| $C_{15}H_{24}O_9$ | Low | Low | ND | Low | ambient cloud water | L |
| $C_{15}H_{29}NO_9S$ | Med | Low | Low | Med | ambient cloud water | M |
| $C_{15}H_{30}O_6S$ | Med | Low | Low | Med | ambient cloud water | M |
| $C_{16}H_{18}O_6$ | Med | Low | Low | ND | syringol aqSOA | A, I, K, L |
| $C_{16}H_{18}O_7$ | Med | Med | Low | Low | syringol aqSOA | A |
| $C_{16}H_{18}O_9$ | Med | Med | ND | Low | syringol aqSOA | K |
| $C_{16}H_{24}O_{11}S$ | ND | Low | ND | ND | ambient cloud water | L |
| $C_{16}H_{31}NO_9S$ | Med | Low | Low | Med | ambient cloud water | M |
| $C_{17}H_{33}NO_9S$ | Med | Low | Low | Med | ambient cloud water | M |
| $C_{18}H_{12}O_5$ | Low | ND | ND | Low | phenol aqSOA | A |
| $C_{18}H_{14}O_4$ | Low | ND | Low | Low | phenol aqSOA | A |
| $C_{18}H_{26}O_{12}S$ | ND | Low | ND | ND | ambient cloud water | L |
| $C_{18}H_{28}O_{11}S$ | Low | Low | ND | ND | ambient cloud water | L |
| $C_{18}H_{38}O_6S$ | Low | ND | ND | Low | ambient cloud water | M |
| $C_{19}H_{30}O_{12}S$ | Low | ND | ND | ND | ambient cloud water | L |
| $C_{20}H_{14}O_6$ | Low | ND | Low | ND | phenol aqSOA | A |
| $C_{20}H_{16}O_7$ | Med | ND | Low | Low | guaiacol aqSOA | A |
| $C_{20}H_{18}O_6$ | Med | ND | Low | Low | guaiacol aqSOA | A |
| $C_{20}H_{26}O_3$ | Low | ND | Low | ND | oxodehydroabietic acid | F |
| $C_{20}H_{28}O_2$ | ND | ND | Low | ND | dehydroabietic acid | F |
| $C_{21}H_{18}O_8$ | Med | Low | Low | Low | guaiacol aqSOA | A |
| $C_{21}H_{20}O_6$ | Low | ND | Low | Low | guaiacol aqSOA | A, K |
| $C_{21}H_{20}O_8$ | Med | ND | Low | ND | guaiacol aqSOA | A |
| $C_{28}H_{26}O_8$ | ND | ND | Low | Low | guaiacol aqSOA | A |

*References: (A) Yu et al. (2016); (B) Desyaterik et al. (2013); (C) Pietrogrande et al. (2014a); (D) Pietrogrande et al. (2014b); (E) Pietrogrande et al. (2015); (F) Mazzoleni et al. (2007); (G) He et al. (2014); (H) Kitanovski et al. (2012); (I) Dzepina et al. (2015); (J) Herrmann et al. (2015); (K) Yu et al. (2014); (L) Cook et al. (2017); (M) Zhao et al. (2013); (N) Hawkins et al. (2018); (O) Lin et al. (2015) and (P) Nguyen et al. (2013).

Field Code Changed
Field Code Changed
Field Code Changed
Field Code Changed
Field Code Changed